# A niobium and tantalum co-doped perovskite cathode for solid oxide fuel cells operating below 500 °C

Mengran Li[1], Mingwen Zhao[2], Feng Li[2], Wei Zhou[3], Vanessa K. Peterson[4], Xiaoyong Xu[1], Zongping Shao[3], Ian Gentle[5] & Zhonghua Zhu[1]

The slow activity of cathode materials is one of the most significant barriers to realizing the operation of solid oxide fuel cells below 500 °C. Here we report a niobium and tantalum co-substituted perovskite $SrCo_{0.8}Nb_{0.1}Ta_{0.1}O_{3-\delta}$ as a cathode, which exhibits high electro-activity. This cathode has an area-specific polarization resistance as low as $\sim 0.16$ and $\sim 0.68\,\Omega\,cm^2$ in a symmetrical cell and peak power densities of 1.2 and 0.7 $W\,cm^{-2}$ in a $Gd_{0.1}Ce_{0.9}O_{1.95}$-based anode-supported fuel cell at 500 and 450 °C, respectively. The high performance is attributed to an optimal balance of oxygen vacancies, ionic mobility and surface electron transfer as promoted by the synergistic effects of the niobium and tantalum. This work also points to an effective strategy in the design of cathodes for low-temperature solid oxide fuel cells.

[1] School of Chemical Engineering, The University of Queensland, St Lucia, Queensland 4072, Australia. [2] School of Physics and State Key Laboratory of Crystal Materials, Shandong University, Jinan 250100, Shandong, China. [3] Jiangsu National Synergetic Innovation Center for Advanced Materials (SICAM), State Key Laboratory of Materials-Oriented Chemical Engineering, College of Chemical Engineering, Nanjing Tech University, No. 5 Xin Mofan Road, Nanjing 210009, Jiangsu, China. [4] Australian Centre for Neutron Scattering, Australian Nuclear Science and Technology Organisation, Lucas Heights, New South Wales 2234, Australia. [5] School of Chemistry and Molecular Biosciences, The University of Queensland, St Lucia, Queensland 4072, Australia. Correspondence and requests for materials should be addressed to W.Z. (email: zhouwei1982@njtech.edu.cn) or to Z.Z. (email: z.zhu@uq.edu.au).

A low-temperature solid oxide fuel cell (LT-SOFC) is a durable energy device that can be deployed to convert the chemical energy stored in various types of fuels into electricity with high efficiency, ease of sealing, and reduced system and operational costs[1–3]. However, the low operating temperature (450-600 °C) typically leads to sluggish kinetics of the oxygen reduction reaction (ORR) at the cathode, with this being a major limitation to LT-SOFC performance[4–9].

Intensive research has been carried out in an effort to explore cathode compositions suitable for operation at low temperature[4,6,7,10–15]. Oxides offering high mixed ionic and electronic conductivities (MIECs) are considered to be some of the most promising candidates for the next generation of SOFC cathodes due to their extended active sites for ORR when compared with purely electronic conducting materials[16,17]. Some of these MIEC cathodes have been reported exhibiting relatively low cathode polarization resistance below 600 °C (ref. 11). For example, the *in-situ* co-assembly of $La_{0.8}Sr_{0.2}MnO_3$ (with a very low $O_2$ dissociation energy barrier) and $Bi_{1.6}Er_{0.4}O_3$ (with fast oxygen incorporation kinetics) leads to a high performance nanocomposite cathode showing a low polarization resistance of $\sim 0.078\,\Omega\,cm^2$ and $\sim 0.6\,\Omega\,cm^2$ at 600 and 500 °C, respectively[11]. Choi *et al.*[12] developed a novel MIEC cathode $PrBa_{0.5}Sr_{0.5}Co_{1.5}Fe_{0.5}O_{5+\delta}$ that exhibits a polarization resistance as low as $\sim 0.33\,\Omega\,cm^2$ at 500 °C, and the $NdBa_{0.75}Ca_{0.25}Co_2O_{5+\delta}$ material also shows an outstanding ORR activity at reduced temperature[7]. Another MIEC cathode composition, $Ba_{0.9}Co_{0.7}Fe_{0.2}Mo_{0.1}O_{3-\delta}$, was also reported to show an enhanced cathode performance with a polarization resistance of $\sim 0.28\,\Omega\,cm^2$ at 500 °C (ref. 18).

Currently, some of the most popular MIEC cathode materials are the stabilized $SrCoO_{3-\delta}$ (SC) perovskite oxides, such as $Sm_{0.5}Sr_{0.5}CoO_{3-\delta}$ (ref. 19), $(La,Sr)(Co,Fe)O_{3-\delta}$ (refs 20,21) and $Ba_{0.5}Sr_{0.5}Co_{0.8}Fe_{0.2}O_{3-\delta}$[4,22], which are claimed to exhibit high ORR activity in the intermediate temperature range 600–750 °C because of their relatively high mixed conductivities[23,24]. The perovskite structure of SC, which is favoured for ORR, is usually stabilized by partial *B*-site substitution with high oxidation-state cations[25], such as $Nb^{26,27}$, $Mo^{28}$, $Sb^{29,30}$ and $P^{31,32}$, and these cations lead to low area-specific resistances (ASRs) at reduced temperature[27–29,31,33,34]. Besides the single doped SCs, Zhou *et al.*[10] developed a highly active perovskite cathode material, featuring a partial replacement of Co ions with both $Sc^{3+}$ and $Nb^{5+}$, and these dopants induce a remarkably high ORR activity at 550 °C. To the best of our knowledge, few studies report the possible synergistic effects of co-doping highly charged dopants on catalysing the ORR in LT-SOFC cathodes.

Herein, we report the study of the synergistic effects of two highly charged *B*-site dopants on the performance of the perovskite LT-SOFC cathode $SrCo_{0.8}Nb_{0.1}Ta_{0.1}O_{3-\delta}$ (SCNT), with this cathode exhibiting outstanding and stable electrochemical performance below 500 °C. A low ASR of $\sim 0.16$ and $\sim 0.68\,\Omega\,cm^2$ is achieved at 500 and 450 °C, respectively, by the SCNT cathode in a symmetrical cell configuration under open circuit conditions. A LT-SOFC with a pure SCNT cathode exhibits good performance of $\sim 1.2$ and $\sim 0.7\,W\,cm^{-2}$ at 500 and 450 °C, respectively. Our results show that the co-substitution of $Nb^{5+}$ and $Ta^{5+}$ can lead to an optimized balance of oxygen vacancies, ionic mobility and surface electron-transfer, which potentially benefit the ORR in the SCNT cathode.

## Results

**Structure and cation arrangement of SCNT.** Joint Rietveld analysis of neutron and X-ray powder diffraction data (Fig. 1a,b) revealed that the SCNT at room temperature exhibits a

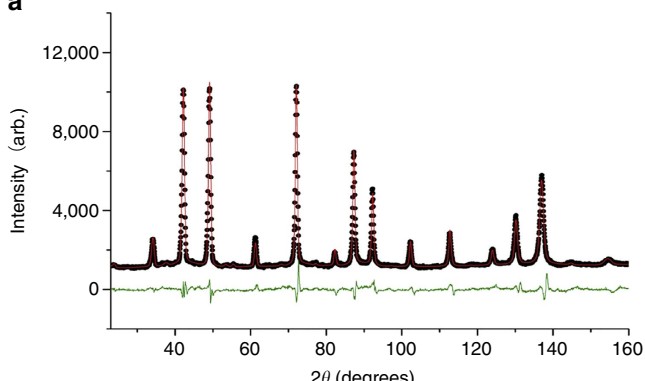

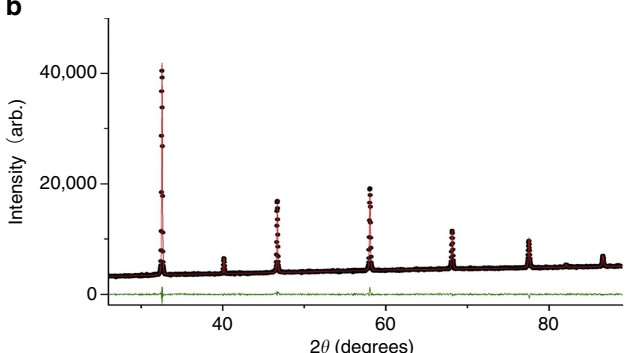

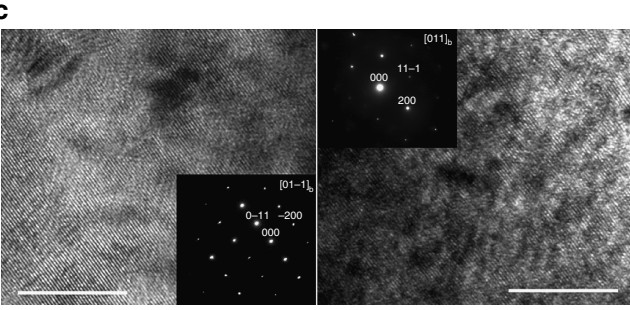

**Figure 1 | Crystal structure analyses of $SrCo_{0.8}Nb_{0.1}Ta_{0.1}O_{3-\delta}$ (SCNT) at room temperature.** Joint Rietveld refinement plot of SCNT powders at room temperature using both neutron powder diffraction (**a**) and X-ray powder diffraction data (**b**). Data are shown as black dots, the calculation as a red line, and the difference between these two as a green line. (**c**) High-resolution transmission electron microscopy bright field images of SCNT with selected area electron diffraction are shown as insets, along the [01 − 1] direction on the left and the [011] direction on the right. Scale bar, 10 nm.

**Table 1 | Crystallographic details of SCNT obtained from joint Rietveld refinement against both neutron and X-ray powder diffraction data.**

| Atom | Site | $x$ | $y$ | $z$ | Occupancy | $U_{iso}$ (Å²) |
|---|---|---|---|---|---|---|
| Sr | 1*b* | 0.5 | 0.5 | 0.5 | 1.000 | 0.012(7) |
| Co | 1*a* | 0 | 0 | 0 | 0.831 (4) | 0.01 |
| Nb | 1*a* | 0 | 0 | 0 | 0.097 (5) | 0.01 |
| Ta | 1*a* | 0 | 0 | 0 | 0.069 (5) | 0.01 |
| O | 3*d* | 0.5 | 0 | 0 | 0.944 (5) | 0.0278(3) |

$a = 3.9066(1)$ Å, $wR = 2.44\%$ , Reduced $\chi^2 = 1.76$.

cubic perovskite structure with $Pm\bar{3}m$ space-group symmetry and a lattice constant of 3.9066(1) Å (Table 1). High-resolution transmission electron microscopy combined with selected area electron diffraction (SAED) (Fig. 1c) confirms this structure. Moreover, the binding energies of Nb 3d 5/2 (206.76 eV) and Ta 4f 7/2 (25.58 eV) in SCNT, obtained from X-ray photo-electron spectroscopy, indicate that the dopants are both in 5+ valence[35,36] (Supplementary Fig. 1). The cubic structure of SC is maintained by the co-doping of $Nb^{5+}$ and $Ta^{5+}$ at the Co-site likely because of their high oxidation states[25]. Rietveld refinement results show Nb and Ta cation doping levels of 9.7(5) and 6.9(5) mol%, respectively, and an oxygen deficiency level of 5.6(5) mol% in SCNT. Both the cubic perovskite structure and oxygen deficiency are beneficial for oxygen-ion conduction, which is critical for a cathode, particularly for LT-SOFC application. The former makes oxygen vacancies migrate freely among lattice-equivalent oxygen sites[37], while the latter facilitates ionic conduction[16,38].

**ORR activity in symmetrical and single cells.** We determined the ORR activity of SCNT in a symmetrical cell configuration between 450 and 700 °C using electrochemical impedance spectroscopy (EIS). The cathode ASR, calculated from the intercept difference of EIS impedance with the real axis (that is, Re_Z in Fig. 2c), is the key variable characterizing the cathode performance, with low ASR indicating high activity. The intercept of the impedance at high frequencies indicates an ohmic resistance arising from the electrolyte, electrode and connection wires[39], with only approximately 1–2% of the total ohmic resistance contributed from the SCNT cathode on both sides of the electrolyte $Gd_{0.1}Ce_{0.9}O_{1.95}$ (GDC)-based symmetrical cell (Supplementary Fig. 2a). The compatibility of SCNT with $Sm_{0.2}Ce_{0.8}O_{1.9}$ (SDC) and GDC electrolytes was examined by comparing the X-ray diffraction patterns of a 50:50 wt.% powder mixture of the SCNT and electrolyte after heating to the cathode fabrication temperature of 1,000 °C for 2 h (Supplementary Fig. 3). The results reveal no obvious changes to the SCNT after heating with electrolyte, indicating a good chemical compatibility between the two. Since the silver current collector does not significantly affect cathode performance[40] and the cathode thickness ($\sim 10\,\mu m$) proves to be sufficient (Supplementary Fig. 2b), our measured ASRs reflect the ORR activity of the SCNT cathode. Figure 2a shows that the

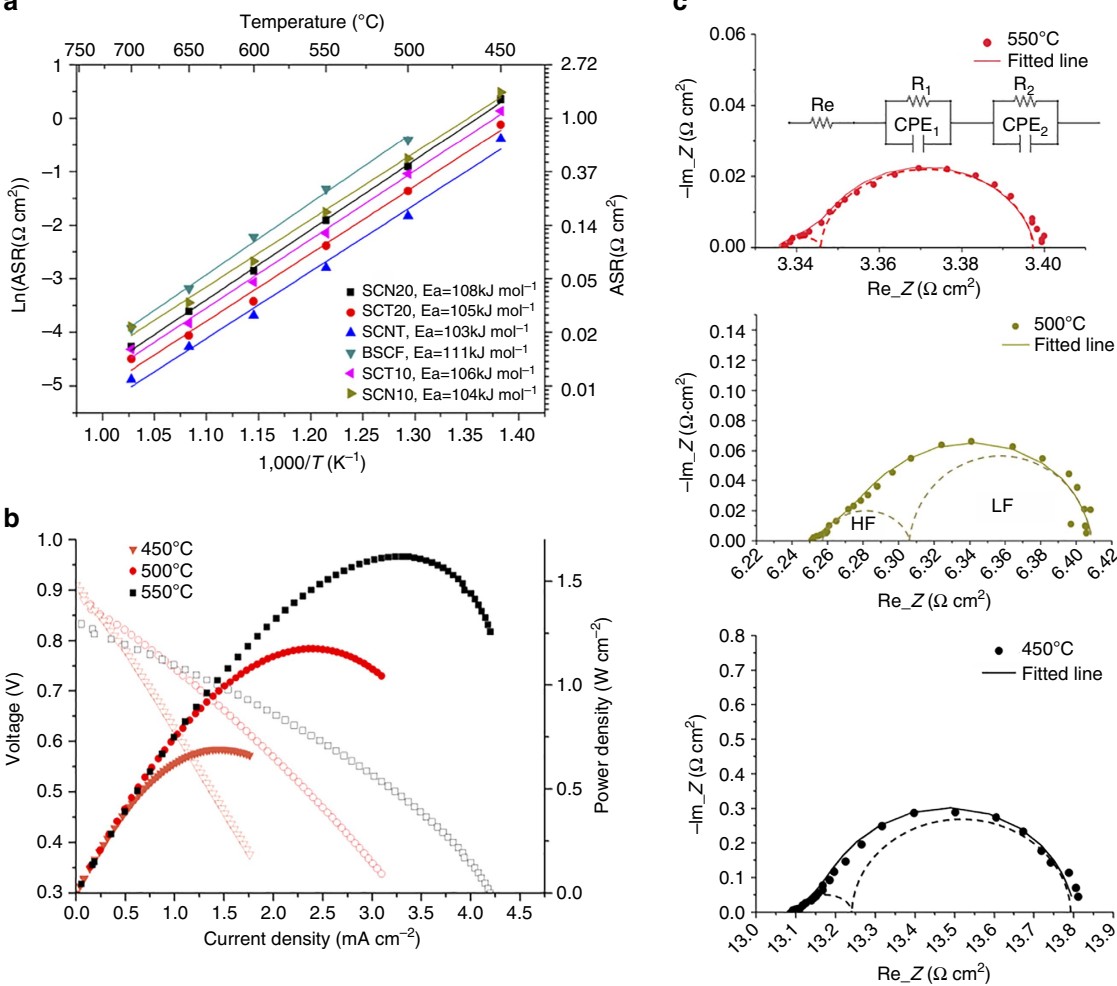

**Figure 2 | Cathode performance evaluation for the SCNT perovskite.** (**a**) Thermal evolution of the ASR of SCNT, $SrCo_{0.9}Nb_{0.1}O_{3-\delta}$ (SCN10), $SrCo_{0.9}Ta_{0.1}O_{3-\delta}$ (SCT10), $SrCo_{0.8}Nb_{0.2}O_{3-\delta}$ (SCN20), $SrCo_{0.8}Ta_{0.2}O_{3-\delta}$ (SCT20) and $Ba_{0.5}Sr_{0.5}Co_{0.8}Fe_{0.2}O_{3-\delta}$ (BSCF) cathodes as prepared and studied under the same conditions. Electrochemical impedance spectroscopy (EIS) results using a $Sm_{0.2}Ce_{0.8}O_{1.9}$ (SDC)-based symmetrical cell. (**b**) Performance of an anode-supported SCNT | GDC($\sim 14\,\mu m$) | GDC + Ni single cell at 450, 500 and 550 °C with $H_2$ at the anode and flowing air at the cathode. (**c**) Example Nyquist plots for the SCNT symmetrical cell and the corresponding fitted impedance spectra using a two-process equivalent circuit model.

SCNT cathode exhibits notably high ORR activity at low temperature, with an ASR as low as 0.061–0.086, 0.16–0.23 and 0.68–0.80 $\Omega$ cm$^2$ at 550, 500 and 450 °C, respectively. The SCNT cathode outperforms the other reported cathode compositions at below 500 °C. (Supplementary Table 1)[7,10–12,18,40–42] For example, the electroactivity of SCNT cathode is nearly twice that of the highly active $SrSc_{0.175}Nb_{0.025}Co_{0.8}O_{3-\delta}$ at 500 °C (ref. 10), and is also higher than that of $Ba_{0.9}Co_{0.7}Fe_{0.2}Mo_{0.1}O_{3-\delta}$ at 450 °C (ref. 18).

When examined against other cathodes, the SCNT cathode performance is also found to be higher than that of the isostructural $SrCo_{0.9}Nb_{0.1}O_{3-\delta}$ (SCN10), $SrCo_{0.9}Ta_{0.1}O_{3-\delta}$ (SCT10), $SrCo_{0.8}Nb_{0.2}O_{3-\delta}$ (SCN20) and $SrCo_{0.8}Ta_{0.2}O_{3-\delta}$ (SCT20) perovskite cathode materials (Supplementary Fig. 4), having ASRs of 0.476 ± 0.009, 0.353 ± 0.001, 0.63 ± 0.08 (ref. 43) and 0.25 ± 0.02 $\Omega$ cm$^2$ (ref. 43), respectively, at 500 °C. In addition, a lower activation energy (103.1 ± 0.8 kJ mol$^{-1}$) of SCNT is observed relative to that of SCN10 (105.3 ± 1.6 kJ mol$^{-1}$), SCT10 (105.3 ± 0.5 kJ mol$^{-1}$), SCN20 (108.5 ± 0.3 kJ mol$^{-1}$) and SCT20 (105.8 ± 1.5 kJ mol$^{-1}$), implying its suitability for catalysing oxygen reduction at low temperature.

The performance of the SCNT cathode in a LT-SOFC was evaluated using Ni-SDC|SDC ($\sim$20 µm) |SCNT ($\sim$10 µm; Supplementary Fig. 5a) and Ni-GDC|GDC ($\sim$14 µm)| SCNT ($\sim$10 µm) fuel cells (Fig. 2b). The micrographs of the two single-cell cross sections are shown in Supplementary Fig. 6. At 550, 500 and 450 °C, power densities of 1.13, 0.77 and 0.37 W cm$^{-2}$ are achieved, respectively, in the former single cell (using SDC as the electrolyte) with ohmic resistances of $\sim$0.072, 0.113 and 0.193 $\Omega$ cm$^2$, which mainly arise from the electrolyte. The electrode polarization resistance (the sum of cathode and anode ASRs) are $\sim$0.059, 0.132 and 0.271 $\Omega$ cm$^2$ at the respective temperature. Given that SCNT has reasonable chemical compatibility with GDC (Supplementary Fig. 3b) and a similar ORR activity with both GDC and SDC electrolyte (Supplementary Fig. 2c), GDC was also used in single cells due to its ease of coating. The cell can generate a peak power density as high as 1.75, 1.22 and 0.7 W cm$^{-2}$ at 550, 500 and 450 °C, respectively, this being significantly higher than that of $Ba_{0.5}Sr_{0.5}Co_{0.8}Fe_{0.2}O_{3-\delta}$ of $\sim$0.97, 0.52 and 0.316 W cm$^{-2}$, respectively (Supplementary Fig. 5b). With a thinner GDC electrolyte, the fuel cell ohmic resistance is reduced to 0.033, 0.049 and 0.083 $\Omega$ cm$^2$ at these temperatures, less than half of that for the SDC ($\sim$20 µm)-based fuel cell. However, the electrode resistance of the GDC cell is only slightly lower than that of the SDC-based cell, being 0.056, 0.116 and 0.242 $\Omega$ cm$^2$ at these respective temperatures. Taking into consideration the ease and low-cost of the ceramic fabrication processes involved in the necessary scale-up[5], GDC electrolyte fuel cells can be fabricated to a thickness of approximately 10–14 µm, though further reduction in GDC thickness is expected to boost the single cell performance by lowering its ohmic resistance[6,9]. Overall, the performance of the SCNT-based fuel cell surpasses the target of 500 mW cm$^{-2}$ for SOFCs[44], suggesting the possibility of practical operation even below 450 °C.

**Synergistic effects of Nb and Ta on the ORR.** Notably, SCNT shows a higher ORR activity when compared with the iso-structural SCN20 and SCT20 materials with similar lattice constants of 3.9066(1) Å for SCNT (Table 1), 3.8978(2) Å for SCT20 and 3.8971(1) Å for SCN20, obtained from the analysis of the neutron powder diffraction (NPD) data in our previous work[43]. The oxygen vacancy content of the SCN20, SCT20 and SCNT materials was also determined from NPD data to be 0.102 ± 0.02, 0.159 ± 0.15 and 0.168 ± 0.15, respectively, revealing a similar oxygen vacancy level in SCNT and SCT20, which are both significantly higher than in SCN20. Thermal gravimetric analysis also shows higher oxygen vacancy contents in SCNT and SCT20 than in SCN20 at elevated temperature. (Supplementary Fig. 7) Given the fixed valence of dopants, the valence of reducible Co is likely the main reason for the oxygen vacancy concentration differences, and the average valence of cobalt in samples was calculated from the elemental composition determined from the structural refinement. The average valence of Co is 3.44, 3.33 and 3.41 for SCN20, SCT20 and SCNT, respectively. The lower Co valence in Ta-doped samples can be ascribed to the lower electronegativity of Ta than Nb[43]. In addition, our first-principles calculation results also show that oxygen formation energies are 1.539, 1.456 and 1.512 eV for the Nb-, Ta- and co-doped models, respectively, which further supports the observed relatively high oxygen deficiency in SCNT as induced by Ta. However, it seems insufficient to explain the better performance of SCNT than SCT20 merely by their oxygen vacancy contents. Given the similar particle size (Supplementary Fig. 8) but slightly lower electrical conductivity (Supplementary Fig. 9) of SCNT compared to SCN20 and SCT20, the superior cathode electroactivity of SCNT is likely attributable to other ORR-related properties such

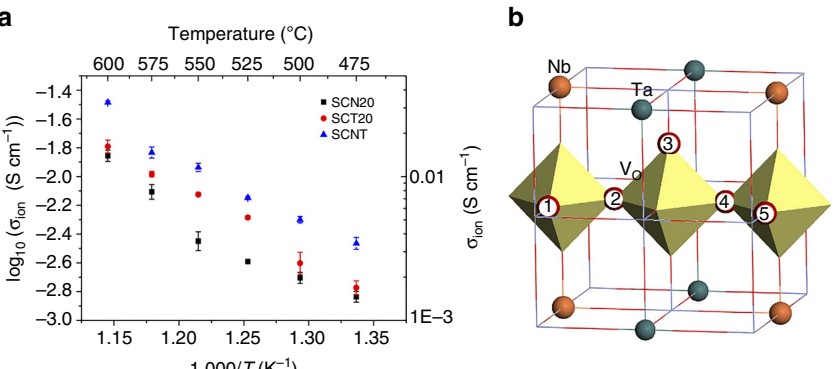

**Figure 3 | Oxygen ionic conductivity study on isostructural single-doped or co-doped $SrCoO_{3-\delta}$ perovskites. (a)** Estimated ionic conductivities of SCN20, SCT20 and SCNT membranes with similar dimensions determined by oxygen permeability testing. The values in the figure are average of multiple data values, and the vertical error bars are estimated from s.d. in the mean. **(b)** A schematic of the minimum energy migration pathway for an oxygen vacancy ($V_O$) in $SrCo_{0.75}Nb_{0.125}Ta_{0.125}O_{3-\delta}$, where dopants shown by coloured balls and Co along the pathway are within the octahedra. Other Co and Sr ions are omitted for clarity. The numbers 1–5 indicate the sequential positions of an oxygen vacancy along the diffusion pathway.

as bulk oxygen ionic conductivity and oxygen surface-exchange kinetics. Hence we estimated the ionic conductivity of the SCN20, SCT20 and SCNT materials by studying the oxygen permeability of dense membranes with similar dimensions from 600 to 475 °C. The higher ionic conductivity (Fig. 3a) of SCNT over SCN20 can be explained by the more oxygen vacancies in SCNT relative to SCN20. Ionic conductivity is also known to be significantly affected by other factors such as lattice geometry, critical radius[45], and lattice free-volume available for oxygen ions to pass through[46]. Because SCNT and SCT20 have similar lattice dimensions, the faster ionic conduction in SCNT may stem from the synergistic effects of Nb and Ta, which potentially decrease the energy barrier for oxygen migration between neighbouring octahedral $CoO_6$ vacancies, as reported for the $Sc^{3+}$ and $Nb^{5+}$-doped perovskite oxide by Zhou et al.[10] In order to confirm this hypothesis, we investigated the pathways for oxygen vacancy migration through first-principles calculations. We found that our three models have very similar minimum energy pathway, as shown in Fig. 3b, but with different energy barriers. The highest energy barrier along the pathway is 0.433, 0.638 and 0.572 eV for Nb-, Ta- and the co-doped models, respectively, (Supplementary Table 2), indicating an easier vacancy mobility within the co-doped model as compared to Ta-doped model. Although SCNT and SCT20 have similar oxygen vacancy levels, the higher ionic conductivity of SCNT than SCT20 is likely a result of the incorporation of Nb that enhances ionic mobility in the lattice.

Additionally, slightly lower electrical conductivity ($\sigma_{total}$) that includes both electronic and ionic conductivity ($\sigma_{ion}$) is observed for SCNT than for SCN20 and SCT20 (Supplementary Fig. 9). The lower electrical conductivity of SCNT is caused by increased oxygen vacancies that can diminish the charge carriers for hopping process. The ionic transference number ($t_{ion} = \sigma_{ion} \times \sigma_{total}^{-1}$) of SCNT, SCN20 and SCT20 was calculated and shown in Supplementary Fig. 10. As the electronic conductivity dominates, the ionic transference number is very small in the studied temperature range. Nevertheless, SCNT has a larger ionic transference number than SCN20 and SCT20; e.g. SCNT has a transference number of $\sim 1.33 \times 10^{-5}$ at 500 °C, which is $\sim 2.7$ and $\sim 2.1$ times that of SCN20 and SCT20, respectively. By extending the oxygen reduction active region and enhancing the ORR kinetics[47,48], the higher oxygen vacancy content and improved mobility of SCNT imparted by the co-doping are likely to be more important than the electronic conductivity to the outstanding ORR performance of SCNT.

The oxygen surface exchange kinetics were investigated by comparing the $O_2$-intake time of each sample in response to an atmosphere change from $N_2$ to air at 500 °C. The SCNT mass equilibrates faster ($\sim 188$ s) than SCN20 ($\sim 245$ s) and SCT20 ($\sim 217$ s), suggesting a faster oxygen surface exchange of SCNT at lower temperature (Supplementary Fig. 11). Therefore, the Nb and Ta together could also synergistically enhance the surface-exchange processes by creating more oxygen vacancies and improving ionic mobility.

We also fitted the impedance spectra of SCNT, SCN20 and SCT20 cathodes to an equivalent circuit model consisting of two dominant reaction processes, with an example fitting presented in Fig. 2c. As shown in Supplementary Fig. 12 and from our previous work[43], the cathode reciprocal resistances at high and low frequencies show different oxygen partial pressure dependencies $m$ (as shown in equation (5)): $m$ is $\sim 0.25$ at high frequencies (HF) and $\sim 0.5$ at low frequencies (LF) According to the relationship between $pO_2$ dependencies and rate-determining-steps as discussed by other researchers[49–51], the

polarization resistance at HF is likely related to charge transfer and that at LF to non-charge-transfer steps.

$$\text{Charge transfer process}: O_{2,ads} + 2e' + V_O^{\bullet\bullet} \Leftrightarrow O_o^{\times} \qquad (1)$$

Where $O_{2,ads}$ is an adsorbed oxygen molecule on the cathode surface, $e'$ is an electron, $V_O^{\bullet\bullet}$ is an oxygen vacancy and $O_o^{\times}$ is an oxygen. Table 2 summarizes the polarization resistance of these two processes. SCNT exhibits significantly lower ASRs for the charge-transfer process than either SCN20 or SCT20, and nearly half of the resistance of SCN20 and similar resistance to SCT20 for the non-charge transfer process. The fast kinetics of charge transfer can be partly attributed to the high oxygen vacancy content of SCNT induced by Ta, since oxygen vacancies are shown to play an important role in the charge-transfer process (equation (1)).

On the other hand, since $Nb^{5+}$ and $Ta^{5+}$ are inert to oxygen surface redox-processes due to their fixed valence, Co plays a vital role in catalysing the oxygen reduction. Therefore, we calculated the atomic orbital-resolved electron density of states projected onto the Co atom in Nb, Ta and Nb/Ta co-doped strontium cobalt oxides using first-principles calculations. As shown in the schematic models (Fig. 4c,f for Nb or Ta single-doped models and Fig. 4i for co-doped model), there are two categories of cobalt atoms: one is the nearest neighbour (NN) Co to the dopant, including Co1 and Co2 in the single-doped model, and Co1, Co2 and Co3 for co-doped model; the other is the next-nearest neighbour (NNN) Co to the dopants, including Co3 in the single-doped model and Co4 in the co-doped model. By comparing the electronic states of these Co atoms, we found that the NN-Co atoms have very similar density of states (DOS) near the Fermi level for these three models. For the NNN-Co atoms, the Ta-doped model (Fig. 4e) exhibits only 60% of DOS of the Nb-doped model (Fig. 4b) near the Fermi level, indicating that Nb is more favourable to increasing the DOS of the NNN-Co near the Fermi level. Due to the beneficial effect of Nb, the DOS of the NNN-Co near the Fermi level of the co-doped model (Fig. 4h) is $\sim 98\%$ that of Nb-doped model. The enhanced DOS at the Fermi level can increase the efficiency of electron-transfer to the adsorbed oxygen species $O_{2,ads}$ (ref. 52), and it is therefore likely that the higher DOS of NNN-Co atoms near the Fermi level, as induced by Nb, is the reason for the faster kinetics of charge-transfer in SCNT than in SCT20, despite their similar oxygen vacancy concentration.

Our experimental and calculation results reveal that the superior electroactivity of SCNT is a result of a balance of the oxygen vacancy content, oxygen-ion mobility and electron-transfer to $O_{2,ads}$, which are imparted by co-doping Nb and Ta.

**Stability tests.** The durability of the cathode was investigated in both symmetrical and single cell configurations, as shown in Fig. 5. The ASR of SCNT in a symmetrical cell configuration was tested under the open circuit condition for $\sim 200$ h. The ORR activity was relatively stable at $\sim 0.033 \, \Omega \, cm^2$ with an ASR increase of $\sim 0.06\%$ per hour during the testing period. The slight increase of the ASR during the stability test is likely to arise from the densification and reduced porosity of the silver current collector during this testing timeframe, which degrades the overall cathode performance[53–55]. Another short-term stability evaluation of the SCNT cathode in a single-cell configuration with $\sim 20 \, \mu m$-thick SDC electrolyte also showed that the SCNT is stable under 0.7 V polarization for at least 150 h at 450 °C in air. The low current density noted in the stability testing arises from the electrolyte thickness, which leads to high ohmic resistance. This stable ORR activity of SCNT is expected given its stable perovskite lattice (Supplementary Fig. 13).

**Table 2 | Comparison of the ASR at both low frequency (LF) and high frequency (HF) for SCNT, SCT20 and SCN20, and those estimated from impedance spectra in a symmetrical cell in flowing air using an equivalent circuit model with two processes.**

| Temperature (°C) | $ASR_{HF}$ ($\Omega\,cm^2$) | | | $ASR_{LF}$ ($\Omega\,cm^2$) | | |
|---|---|---|---|---|---|---|
| | **SCNT** | **SCT20** | **SCN20** | **SCNT** | **SCT20** | **SCN20** |
| 450 | 0.14 (7) | 0.40 (4) | 0.62 (1) | 0.53 (7) | 0.50 (4) | 1.57 (1) |
| 500 | 0.05 (3) | 0.12 (1) | 0.149 (2) | 0.11 (2) | 0.13 (1) | 0.400 (2) |
| 550 | 0.007 (7) | 0.036 (8) | 0.057 (1) | 0.054 (1) | 0.057 (1) | 0.123 (1) |
| 600 | 0.003 (2) | 0.014 (6) | 0.021 (1) | 0.022 (3) | 0.020 (8) | 0.063 (1) |
| 650 | 0.002 (2) | 0.007 (3) | 0.016 (1) | 0.012 (5) | 0.010 (4) | 0.021 (1) |

ASR, area-specific resistance; SCNT, $SrCo_{0.8}Nb_{0.1}Ta_{0.1}O_{3-\delta}$.

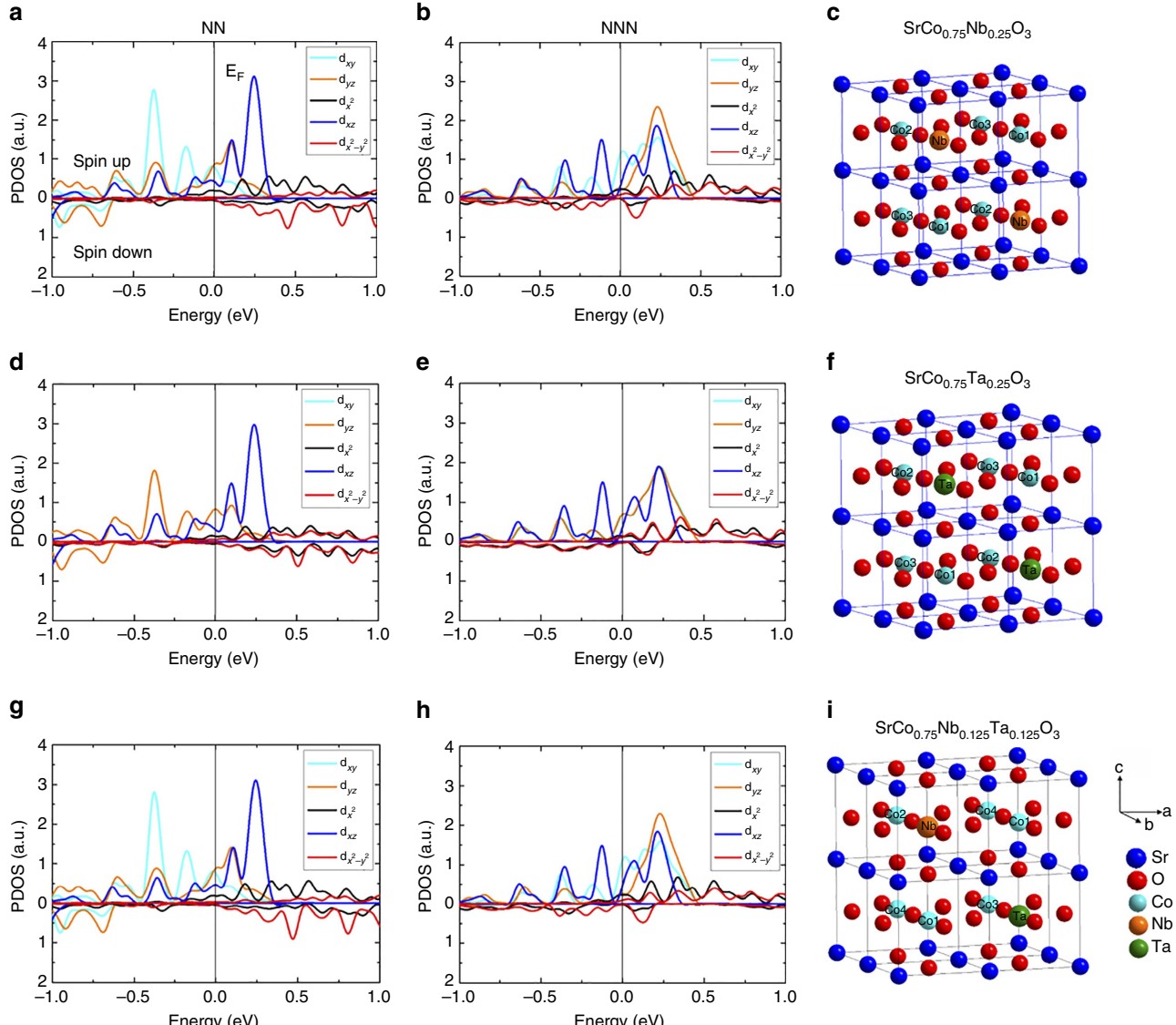

**Figure 4 | Density of states of the neighbouring Co atoms to the dopants in single-doped and co-doped models.** Atomic-orbital-resolved electron density of states (PDOS) projected onto the nearest neighbouring (NN) Co atoms (left column) and the next nearest neighbour (NNN) Co atoms (middle column) of (**a,b**) $SrCo_{0.75}Nb_{0.25}O_{3-\delta}$, (**d,e**) $SrCo_{0.75}Ta_{0.25}O_{3-\delta}$ and (**g,h**) $SrCo_{0.75}Nb_{0.125}Ta_{0.125}O_{3-\delta}$ perovskite oxides, and the corresponding schematic of unit cells (right column). The energy at the Fermi level is set to zero. (**c,f,i**) A schematic of the corresponding single and co-doped models for the first-principles calculations.

## Discussion

In summary, the perovskite composition $SrCo_{0.8}Nb_{0.1}Ta_{0.1}O_{3-\delta}$ (SCNT) was synthesized and exhibits the highest reported activity for the reduction of oxygen in an LT-SOFC, to the best of our knowledge, with an ASR of only $\sim 0.16$ and $\sim 0.68\,\Omega\,cm^2$ at 500 and 450 °C, respectively, in a symmetrical cell configuration. High power density is therefore achieved using a pure SCNT cathode as a result of its outstanding ORR activity.

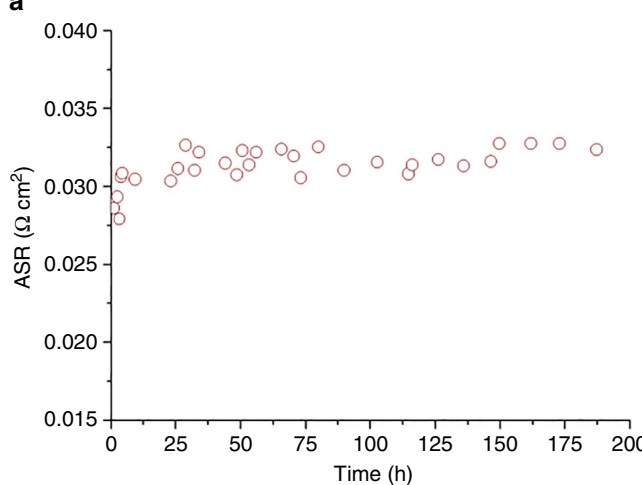

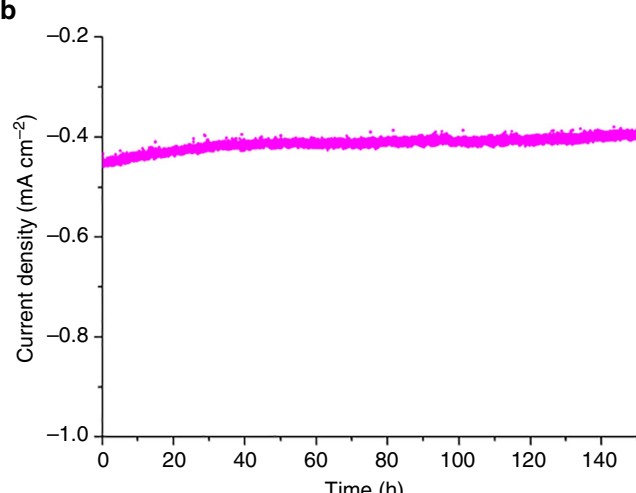

**Figure 5 | Stability test on the SCNT cathode material. (a)** ASRs of SCNT in a symmetrical cell under open-circuit conditions at 600 °C for 200 h **(b)** current density of a SCNT|SDC (~20 μm) | Ni + SDC single cell under 0.7 V polarization in air at 450 °C for 150 h.

A performance comparison amongst the iso-structural SCNT, $SrCo_{0.8}Nb_{0.2}O_{3-\delta}$ (SCN20) and $SrCo_{0.8}Ta_{0.2}O_{3-\delta}$ (SCT20) cathodes reveals enhancement of the bulk oxygen ionic-conductivity achieved through co-doping of $Nb^{5+}$ and $Ta^{5+}$. Our experimental results and density functional theory calculations both show that co-doping results in a favourable balance of oxygen vacancy content, ion mobility and surface electron transfer, which is consistent with the higher performance of the co-doped SCNT cathode at lower temperature. Therefore, our highly active perovskite cathode not only presents a simple solution to address sluggish cathode kinetics below 500 °C, but could also provide an effective doping strategy for the design of mixed-conducting materials for SOFC and oxygen-ion transport membrane applications at low temperature.

## Methods

**Sample syntheses.** The SCNT material was synthesized through a solid state reaction route by ball milling stoichiometric amounts of $SrCO_3$ (≥99.9%, Aldrich), $Co_3O_4$ (≥99.9%, Aldrich), $Nb_2O_5$ (≥99.9%, Aldrich) and $Ta_2O_5$ (≥99.9%, Aldrich) for 24 h, before pelletizing and sintering the mixture in stagnant air at 1,200 °C for 10 h. Subsequently, the sintered pellets were well ground and re-sintered for another 10 h at 1,200 °C. $SrCo_{0.9}Nb_{0.1}O_{3-\delta}$ (SCN10), $SrCo_{0.8}Nb_{0.2}O_{3-\delta}$ (SCN20), $SrCo_{0.9}Ta_{0.1}O_{3-\delta}$ (SCT10), $SrCo_{0.8}Ta_{0.2}O_{3-\delta}$

(SCT20) and $Ba_{0.5}Sr_{0.5}Co_{0.8}Fe_{0.2}O_{3-\delta}$ (BSCF) were also prepared through a similar synthesis route.

**Structure characterization.** The crystal structures of cathode materials were studied by X-ray powder diffraction and NPD. High-resolution NPD data were collected using ECHIDNA, the high-resolution neutron powder diffractometer at the Australian Nuclear Science and Technology Organization (ANSTO)[56], with a neutron wavelength of 1.6219(2) Å, determined using the $La^{11}B_6$ NIST standard reference material 660b. NPD data were obtained from SCNT within a 6 mm vanadium can for 6 h in the $2\theta$ angular range 4 to 164° with a step size of 0.125°. GSAS-II (ref. 57) was employed to perform Rietveld analysis of the high-resolution NPD data, using a $Pm\bar{3}m$ cubic perovskite[33] starting structure. The structure was refined against both the X-ray powder diffraction and NPD data, with atomic displacement parameters for the Co, Nb and Ta, fixed to 0.01. High-resolution electron transmission microscopy (HR-TEM, Tecnai F20) in conjunction with selected area electron diffraction was also used for phase identification.

**Conductivity and thermogravimetric analyses.** A DC 4-probe method was used to measure electrical conductivity of the specimen in flowing air (200 ml min[-1]). The samples for the conductivity measurement were dense bars, which were prepared by pressing the cathode powders followed by sintering at 1,200 °C for 5 h. Following this, samples were well milled and polished and silver leads attached as the current and voltage electrodes. Electrical conductivity was measured using an Autolab PGSTAT20 work station.

Ionic conductivities were estimated from oxygen permeability tests carried out by gas chromatography (GC)[58]. Membranes were fabricated by pelletizing cathode powders (milled for 2 h in alcohol at 400 r.p.m.), followed by sintering at 1,200 °C for 10 h and polishing. The relative densities of all samples were over 95%. Subsequently, the dense pellets were sealed in an alumina tube using silver paste. The effective area of the membranes were ~65 mm² with a thicknesses of 0.07 cm. Helium was applied at one side as the sweep gas with a rate of 100 ml min[-1] and the other side was exposed to air. The overall resistance to oxygen permeation was calculated from the following equation:

$$R_{overall} = \frac{RT}{16F^2}\frac{1}{SJ_{O_2}}\left[\ln\left(\frac{P'_{O_2}}{P''_{O_2}}\right)\right] \tag{2}$$

where

 $R$ = ideal gas constant
 $F$ = Faraday constant
 $S$ = valid area of the membrane
 $J_{O_2}$ = oxygen permeation flux
 $P'_{O_2}$ = oxygen partial pressure at the side of membrane exposed to air
 $P''_{O_2}$ = oxygen partial pressure at the sweep side

It was assumed that bulk ionic conduction dominated the oxygen permeation process because of the relative thickness of the membranes, and therefore $R_{overall}$ is roughly equal to $R_{ionic}$. Hence, the ionic conductivity of the sample was estimated according the following equation:

$$\sigma_{ionic} = \frac{1}{R_{ionic}} \times \frac{S}{L} \approx \frac{1}{R_{overall}} \times \frac{S}{L} \tag{3}$$

Where $L$ = the thickness of the membrane.

The ionic transference ($t_{ion}$) of the samples were calculated using the following equation:

$$t_{ion} = \frac{\sigma_{ionic}}{\sigma_{total}} \tag{4}$$

Where $\sigma_{total}$ and $\sigma_{ion}$ are the total electrical and ionic conductivity, respectively.

Thermal gravimetric analysis was performed by PerkinElmer STA6000 to monitor the mass change of SCNT, SCT20 and SCN20 in flowing air from 200 to 800 °C and also during the abrupt change of atmosphere from flowing air to nitrogen to air at 500 °C. Specimens were pelletized and ground using a mortar and pestle to ensure similar grain size before the test. Samples were first gradually heated to 200 °C and held for 1 h to remove absorbed moisture. The temperature was then increased at a rate of 1 °C min[-1] to 500 °C in flowing air (20 ml min[-1]). Subsequently, the flowing gas was abruptly switched to nitrogen, and this condition remained for 2 h until the sample weight stabilized. Then, the atmosphere was switched back to air and the mass change recorded until equilibrium was reached. The rate of weight change was estimated by:

$$rate_{mass\_change} = \frac{m_{t+\Delta t} - m_t}{\Delta t} \tag{5}$$

Where $m_t$ is the weight of the sample at time $t$, $\Delta t$ is the time interval between two recorded adjacent points.

**ORR characterization.** Cathode polarization resistance was characterized in a cathode|ceria-based electrolyte|cathode symmetrical cell configuration using electrochemical impedance spectroscopy (EIS) carried out with an Autolab PGSTAT20. The samples were measured at least three times to ensure accuracy.

The ceria-based electrolyte was either $Sm_{0.2}Ce_{0.8}O_{1.9}$ (SDC, from Fuel Cell Materials) or $Gd_{0.1}Ce_{0.9}O_{1.95}$ (GDC, from Aldrich). The symmetrical cells were fabricated by spraying nitrogen-borne cathode slurries onto both sides of SDC dense disks, followed by calcination at 1,000 °C for 2 h in stagnant air. Cathode slurries were prepared by suspending powder cathodes in isopropyl alcohol. The thicknesses of cathodes were controlled to be around 10 μm, and the active area of each cathode was ∼1.15 cm². SCNT cathodes with different thicknesses were also fabricated by changing the spraying time. Silver paste was subsequently painted onto both cathode sides as current collector. The symmetrical cell with a silver electrode was fabricated by painting the silver paste onto both sides of the GDC disk, followed by baking at 260 °C for 30 min.

We evaluated the performance of the LT-SOFC using anode-supported button-like single cells. The anode powders were prepared by ball milling the NiO, GDC or SDC, and dextrin (pore former) with a weight ratio of 6:4:1 for 20 h in ethanol. The anode-supported single cells were fabricated by drop coating the GDC slurry onto the surface of the anode disks, which were fabricated by pressing anode powders into disks and sintering at 900 °C for 5 h. The GDC slurry used in drop coating was prepared by suspending the GDC powders in terpineol and ethanol. The coated disks were subsequently sintered at 1,400 °C for 5 h. The fuel cell for SDC-based cell stability test was fabricated using co-press method[4]. The cathode fabrication was carried out following similar steps to those for producing the symmetrical cell. The mechanisms of the SCNT ORR were studied by fitting the EIS impedance spectra at different $pO_2$ to the $R_e$ ($R_1CPE_1$) ($R_2CPE_2$) equivalent circuit model, where by using the LEVM software. The results are presented in (Supplementary Fig. 12) $R_e$ represents the ohmic resistance; ($R_1CPE_1$) and ($R_2CPE_2$) stand for the two ORR steps at high frequency and low frequency respectively. The rate determining steps of ORR are indicated by a parameter $m$ given as follows[50]:

$$\frac{1}{R_p} \propto P_{O_2}^m \quad (6)$$

Where $R_p$ is the polarization resistance of the corresponding ORR process.

**First-principles calculations.** First-principles calculations were performed with the Vienna *ab initio* simulation package (VASP)[59,60] using density-functional theory. Ion-electron interactions were treated using projector-augmented-wave potentials[61] and a generalized gradient approximation (GGA) in the form of Perdew–Burke–Ernzerhof was adopted to describe electron–electron interactions[62]. The GGA + U calculations were performed with the simplified spherically averaged approach applied to d electrons, where the coulomb (U) and exchange (J) parameters are combined into the single parameter $U_{eff}$ ($U_{eff} = U - J$), which was set to 0.8 eV in these calculations. Electron wave functions were expanded using plane waves with an energy cut-off of 520 eV. The Kohn–Sham equation was solved self-consistently with a convergence of $10^{-5}$. The Brillouin zone was sampled using a $3 \times 3 \times 3$ k-point grid. The stoichiometry of the simulated systems was set to $SrCo_{0.75}Nb_{0.25}O_3$, $SrCo_{0.75}Ta_{0.25}O_3$ and $SrCo_{0.75}Nb_{0.125}Ta_{0.125}O_3$, respectively, due to computational limitation, and the Nb and Ta in SCNT are regarded as ordered instead of randomly distributed for simplification. The formation energy of an oxygen vacancy was calculated from the energy difference between the total energy of the $V_O$-containing sample and sum of the total energy of pristine sample and the chemical potential of an oxygen atom in an $O_2$ molecule. The minimum energy pathway for $V_O$ migration was determined using a climbing image nudged band method. (CNBE)[63,64] implemented in the VASP code.

**Data availability.** The data that support the findings of this study are available from the corresponding authors upon reasonable request.

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

## Acknowledgements

We appreciate the technical support from the Centre for Microscopy and Microanalysis at the University of Queensland, and at the Australian Centre for Neutron Scattering at ANSTO. This work is financially supported by Australian Research Council (DP130102151) and Mengran Li acknowledges additional financial support from the scholarship from China Scholarship Council. Professor Zhu acknowledges the Open Funding from State Key Laboratory of Material-Oriented Chemical Engineering (No. KL15-04). This work was financially supported by the National Nature Science Foundation of China under contract No. 21576135 and the Youth Fund in Jiangsu Province under contract No. BK20150945, and the Priority Academic Program Development of Jiangsu Higher Education Institutions (PAPD).

## Author contributions

Z.Z. and W.Z. directed the research projects; M.L. conducted the experiments and summarized the data; V.K.P. conducted the NPD and performed the refinement; M.Z. and F.L. performed the modelling; all authors discussed the results and contributed to the paper.

## Additional information

**Competing financial interests:** The authors declare no competing financial interests.

