## [Peer Review File · Nature Communications]

Reviewers' comments:

Reviewer #1 (Remarks to the Author):

This is a very interesting and important paper in the development of novel cathodes for low temperature solid oxide fuel cells. The authors introduced a new perovskite cathode material SCNT by co-doping the Nb and Ta ions into $\text{SrCoO}_{3-\delta}$ (SC), and claimed that SCNT exhibits by far the highest activity over oxygen reduction below 500 {degree sign}C as the cathode for LT-SOFC application, and studied the synergistic effects of the co-doping through comparison between SCNT and isostructural Nb or Ta doped cathodes. They reported that the outstanding ORR activity of SCNT is mainly due to the synergistic effects of co-substitution, which enhance the ionic conductivity and makes the neighbouring Co ions more active for charge transfer. The paper is well written and I would recommend the acceptance of paper. There are only some minor issues need to be addressed:

1. 1st paragraph, Introduction, Page 3. "...low operating temperature (450{degree sign}C - 650{degree sign}C)". I think 650{degree sign}C is still within the intermediate temperature range.

2. 2nd Paragraph, Introduction, Page 3. "...exhibited relatively low cathode polarization...". "cathode polarization" should be "cathode polarization resistance"

3. Structure and cation arrangement of SCNT, Page 6.

- "...cation doping levels of 9.7(5) % and 6.9(5) %..." and "... 5.6(5) % deficient..." It is recommended to clarify what the percentages based on.

- The authors need to provide evidence to confirm the oxidation states of Nb and Ta, it is possible that Nb or Ta is not in +5 but in +3 or other states for example.

- "The former... ionic conduction" This sentence needs reference to support it.

4. ORR activity in symmetrical and single cells, Page 8.

- "The ASR, as calculated from..." The authors are suggested to provide more details to illustrate how they calculated the ASR from the impedance.

- How many times have the authors repeated the ASR experiments for the accuracy?

- I would like to see the XRD profiles of SCN10 and SCT10 cathode under investigation, since structure is important for ORR activity. Are they also in a perovskite structure?

5. Synergistic effects of Nb and Ta on the ORR.

- Page 10. "...sharing similar lattice constants, with ..." I noticed that the lattice constant of SCNT presented here is a little different from the value given before. Why there is a difference, and how the authors achieve the lattice constants of other samples such as SCN20 and SCT20.

- Page 11. "...the ionic conductivity of the SCN20, SCT20 and SCNT series of cathode materials..." It is advised the "series of cathode materials" is removed.

- It is recommended to provide more discussions to explain the reason why high ionic conductivity is more significant over electronic conductivity for the good cathode performance in the main text.

- Page 12. "...in response to atmosphere change...". What is the atmosphere change? The authors need to provide more details to clarify the discussion.

6. Stability tests.

- Page 15. "...under open circuit voltage" It is better to use "under open circuit condition"

- Figure 5(a). Please comment the possible reasons for the first 4h performance degradation

Reviewer #2 (Remarks to the Author):

In this work, the authors report on the improved activity in low-temperature solid oxide fuel cells (LT-SOFC) by co-doping SrCoO_3 with Nb and Ta, claiming the synergetic effects of co-doping. For a non-expert in LT-SOFC (like me), the authors' claim of "highest activity so far" is not persuasive because they did not provide any comparison with state-of-the-art performance of LT-SOFC. In any case, I pass the judgment of how impressive the present result is to other referees or Editor. Instead, I paid more attentions to the first-principles part in the manuscript. The authors carried

out the first-principles calculations to support the synergetic effects of Nb and Ta co-doping. However, I notice several critical problems in the computational results and ensuing analysis, as follows:

1. The vertical scale in Figure 4(c) is different from those in Figure 4(a) and (b). This makes the enhancement of DOS at the Fermi level difficult to capture from the figure. The authors should use the same scale and provide the detailed number.
2. The authors projected DOS on the Co atoms nearest to dopants. However, there are different types among the nearest Co atoms. For example, in Figure 4(a), dopant-coordination numbers of Co atoms are 0, 2, and 4. Which atoms are used in Figure 4(a) (and also (b) and (c))?
3. In addition, non-neighboring Co atoms should be also discussed in terms of the change in PDOS because they can also be the catalytic sites when exposed at the surface.
4. Most importantly, the increase of DOS by co-doping is difficult to rationalize if oxidation states of Nb and Ta in SCNT are the same as in SCN or SCT. The authors should clarify why DOS in SCNT is not a linear interpolation between those in SCN and SCT. This should be possible with the detailed analysis on the electronic structure.
5. Related to 4, magnetic moments on Nb and Ta sites should be also analyzed if they are not in 5+ state.
6. The computational section lacks in the detailed information on the computation (k-point mesh, U values, etc.)

In conclusion, I do not think the computational analysis in the present manuscript is solid enough to support the synergetic effect of Co-doping. As such, I do not recommend this manuscript to the publication in Nature Communications.

Reviewer #3 (Remarks to the Author):

Manuscript presents a new high performance (low ASR) SCNT cathode for low temperature SOFC operation. This is technology that can have major impact and as such is appropriate for this journal. However, some details cause concern and require significant clarification:

The ASR values were determined by EIS on SDC pellets using Ag current collectors. The ASR values are very low, which is the point of paper but it raises concern over signal to noise. At 550°C (Fig 2 c) non-ohmic part is $\sim 0.06 \text{ Ohm cm}^2$, but total is 3.4 Ohm cm^2 . How accurate were the relative measurements? What was pellet thickness and does the high frequency part correspond to the pellet ASR, or is there an additional Ohmic contribution to the cathode ASR?

ASR depends not only on cathode composition, but also thickness, microstructure, etc. The cathode is supposedly only 10 micron thick which is too thin for real SOFC resulting in large sheet resistance. Ag paste was used to address this for button cell, but then also contributes to cathode performance. How thick was the Ag current collector, and did any of the Ag enter the cathode pores? Fig 3 SEMs don't show Ag coating.

The discussion states a synergy between Nb and Ta on the ORR, but how that is done is not shown. It assumes HF arc due to charge transfer and LF arc due to non-charge transfer, but also not shown. If assumption state so.

Since Nb⁵⁺ and Ta⁵⁺ are fixed valent, how would they contribute to ORR? DOS projections indicate that by co-doping they impose a greater effect on Co, but fact that they substituted for Co means there is less Co if as expect Co is the active cation in ORR.

One of the most important Fig's is S8 which is in Supplemental rather than main text. It shows the conductivity of SCNT compared to the other single doped compounds SCT20 and SCN20. This 4pt measurement should have less error than the EIS and it shows SCNT is LESS conductive than

SCT20 and SCN20 in the temperature range of interest, 400-550{degree sign}C. If SCNT is the better cathode why does it have lower conductivity? I assume answer will be because that is electronic and not ionic conductivity, but then you need ionic conductivity measurements in this temperature region.

O2 permeation measurements were done to separate out ionic conductivity and it shows higher conductivity for SCNT, but only over 700-860{degree sign}C temp range which is not the temperature range of the ASR and SOFC measurements. Also these types of measurements are prone to gas leaks. Did they use a tracer gas to determine any leak rate?

Minor points:

GDC is Gadolinia Doped Ceria, which is "Gd" but authors keep using "Ga" which is Gallium.

Pg 3 reference 11 is cited for their LSM-ESB cathode having high performance below 600{degree sign}C and authors state this is "result of enhanced ionic conduction". However, the ref 11 authors themselves describe this in terms of enhanced ORR due to LSM having excellent O2 dissociation and ESB having excellent O incorporation steps. This should be more accurately reflected in current manuscript as well as including the performance itself which was 0.078 Ohm cm2 at 600{degree sign}C which is comparable to the following ref 12 performance.

Fig 5a shows an increase in ASR with time. What is the % change per hr?

Fig S7 identify the layers. Also, (b) is poor resolution image with charging.

Responses to Reviewers' comments for paper NCOMMS-15-24507A

Reviewer #1

This is a very interesting and important paper in the development of novel cathodes for low temperature solid oxide fuel cells. The authors introduced a new perovskite cathode material SCNT by co-doping the Nb and Ta ions into $\text{SrCoO}_{3-\delta}$ (SC), and claimed that SCNT exhibits by far the highest activity over oxygen reduction below 500 °C as the cathode for LT-SOFC application, and studied the synergistic effects of the co-doping through comparison between SCNT and isostructural Nb or Ta doped cathodes. They reported that the outstanding ORR activity of SCNT is mainly due to the synergistic effects of co-substitution, which enhance the ionic conductivity and makes the neighbouring Co ions more active for charge transfer. The paper is well written and I would recommend the acceptance of paper. There are only some minor issues need to be addressed:

1. 1st paragraph, Introduction, Page 3. "...low operating temperature (450°C - 650°C)". I think 650°C is still within the intermediate temperature range.

Reply:

We would like to appreciate gratefully Reviewer #1's immense time contribution to our manuscript, and revising this manuscript has been a great learning process and pleasure for us.

We agree with Reviewer #1 opinion on the temperature range of LT-SOFCs. Correspondingly, we have corrected the sentence as follows:

"...the low operating temperature (450-600 °C) typically leads to sluggish kinetics ..."

2. 2nd Paragraph, Introduction, Page 3. "...exhibited relatively low cathode polarization...". "cathode polarization" should be "cathode polarization resistance"

Reply:

Reviewer #1 is correct, so we have added the "resistance" to that sentence as shown in below:

"...relatively low cathode polarization resistance below 600 °C..."

3. Structure and cation arrangement of SCNT, Page 6.

- "...cation doping levels of 9.7(5) % and 6.9(5) %..." and "... 5.6(5) % deficient..." It is recommended to clarify what the percentages based on.
- The authors need to provide evidence to confirm the oxidation states of Nb and Ta, it is possible that Nb or Ta is not in +5 but in +3 or other states for example.
- "The former... ionic conduction" This sentence needs reference to support it.

Reply:

Reviewer #1 made a valid point here:

- The content percentage is molar percentage based on per mol of the sample, and we have added some words as highlighted in the main article to address this issue.
- According to Reviewer #1's suggestions, we conducted the X-ray photoelectron analysis on SCNT, and proved that the valence of Nb and Ta are both 5+. The Nb and Ta in the respective

SCN20 and SCT20 samples are all in +5 oxidation state, which has been demonstrated in our previous work.¹

Accordingly, we added a new Supplementary Fig. S2 to demonstrate it, and added a few words in the main article to illustrate it as follows:

“Moreover, the binding energies of Nb 3d5/2 (206.76 eV) and Ta 4f 7/2 (25.58 eV) in SCNT, as shown in X-ray photoelectron spectroscopy (XPS) profile, indicate that the dopants are both in 5+ valence.^{34,35} (Supplementary Fig. S2)”

- Thank Reviewer #1’s advice, and we have cited reference 37, 38 and 39 to support that sentence.

4. ORR activity in symmetrical and single cells, Page 8.

- "The ASR, as calculated from..." The authors are suggested to provide more details to illustrate how they calculated the ASR from the impedance.
- How many times have the authors repeated the ASR experiments for the accuracy?
- I would like to see the XRD profiles of SCN10 and SCT10 cathode under investigation, since structure is important for ORR activity. Are they also in a perovskite structure?

Reply:

- Thanks for the recommendation. In order to clarify it, we have added a few words in the main article as follows:

“The ASR, calculated from the intercept difference of EIS impedance spectra with the real axis, is the key variable characterizing the ORR activity, with low ASR indicating high activity.”

- We measured the ASRs of the samples under investigation at least three times for the accuracy.
- We agree with Reviewer #1’s comments that the structure is important for the ORR, and SCN10 and SCT10 are both in similar perovskite structure with SCNT. Additionally, we have added the XRD profiles of SCN10 and SCT10 in the modified Supplementary Fig. S1.

5. Synergistic effects of Nb and Ta on the ORR.

Page 10. "...sharing similar lattice constants, with ..." I noticed that the lattice constant of SCNT presented here is a little different from the value given before. Why there is a difference, and how the authors achieve the lattice constants of other samples such as SCN20 and SCT20.

Page 11. "...the ionic conductivity of the SCN20, SCT20 and SCNT series of cathode materials..." It is advised the "series of cathode materials" is removed. It is recommended to provide more discussions to explain the reason why high ionic conductivity is more significant over electronic conductivity for the good cathode performance in the main text.

Page 12. "...in response to atmosphere change...". What is the atmosphere change? The authors need to provide more details to clarify the discussion.

Reply:

- We appreciate Reviewer #1's meticulous observation. The slight lattice constant difference arises from different powder diffraction data. We obtained lattice constant for SCNT by joint Rietveld refinement using both XRD and NPD profiles of SCNT, and then we analysed the XRD data of SCNT, SCN20 and SCT20 via Le Bail method in order to compare their lattice constants. It seems to be a little ambiguous to readers, so we decided to use the lattice constants obtained also from the Retveld refinement on NPD data in our previous work to ensure the consistency. Therefore, we deleted the Table S1 in Supplementary Information, and also made a few modifications in the main article as follows:

"...SCN20 and SCT20 materials sharing similar lattice constants, with values of 3.9066(1) Å for SCNT (Table 1) , 3.8978(2) Å for SCT20, and 3.8971(1) Å for SCN20 obtained from the analysis of the NPD in our previous work.⁴³"

- Thanks for Reviewer #1's invaluable advice. We have corrected the language issue.

"...Hence we estimated ionic conductivity of the SCN20, SCT20, and SCNT through studying the oxygen permeability of dense membranes with similar dimensions from 600 to 475 °C"

In addition, we have made a few changes:

"Additionally, slightly lower electrical conductivity, including both electronic and ionic conductivity with the electronic one dominating, is observed for SCNT compared with SCN20 and SCT20 (Supplementary Fig. S10). The lower electrical conductivity is caused by more oxygen vacancies in SCNT that can diminish the charge carriers for hopping process. By extending the oxygen reduction active region and enhancing the ORR kinetics^{39,48}, the higher oxygen vacancy content and improved mobility of SCNT imparted by the co-doping are more important than the electronic conductivity for the outstanding ORR performance of SCNT"

- We agree with Reviewer #1 that we should provide more details in the main article. The atmosphere in the furnace chamber is changed from nitrogen to air, and the following is what we added into the main article for clarification:

"The oxygen surface exchange kinetics were investigated by comparing O₂-intake time of each sample in response to the atmosphere change from N₂ to air at 500 °C."

6. Stability tests.

- Page 15. "...under open circuit voltage" It is better to use "under open circuit condition"
- Figure 5(a). Please comment the possible reasons for the first 4h performance degradation

Reply:

- Reviewer #1 is correct. We have corrected this error. *"... tested under the open circuit condition for..."*
- We feel grateful for Reviewer #1' meticulous observation. The degradation was mainly due to the microstructure growth as well as the porosity reduction of the silver current collector, which has been reported to be the main reason for the cathode performance degradation². Correspondingly, we have added it to the main article:

“The slight increase of ASR during the stability test is likely to arise from the densification and the reduced porosity of the silver current collector, which degrades the overall cathode performance during this testing timeframe.”^{50,51,52} ”

Reviewer #2

In this work, the authors report on the improved activity in low-temperature solid oxide fuel cells (LT-SOFC) by co-doping SrCoO₃ with Nb and Ta, claiming the synergetic effects of co-doping. For a non-expert in LT-SOFC (like me), the authors' claim of "highest activity so far" is not persuasive because they did not provide any comparison with state-of-the-art performance of LT-SOFC. In any case, I pass the judgment of how impressive the present result is to other referees or Editor. Instead, I paid more attentions to the first-principles part in the manuscript.

Reply:

We would like to express our great gratitude to Reviewer #2's invaluable comments that significantly enhance the quality and accuracy of our manuscript.

According to Reviewer #2's opinion on ORR activity, we provided the following table in the Supplementary Information, listing the area specific resistance (ASR) values of other cathode compositions that have been reported to be highly active below 500°C. SCNT exhibits much lower ASR as compared to those cathodes especially below 500°C. Additionally, we have mentioned some of them in the introduction, and compared SCNT cathode performance with some of them in the discussion, which is shown in below:

"The SCNT cathode outperforms the other reported cathode compositions at below 500 °C (Supplementary Table S1)^{7,10,18,40,41,42}. For example, the ORR activity of SCNT cathode is nearly twice that of the highly active ORR catalyst SrSc_{0.175}Nb_{0.025}Co_{0.8}O_{3-δ}¹⁰ at 500°C, and is also higher than that of Ba_{0.9}Co_{0.7}Fe_{0.2}Mo_{0.1}O_{3-δ}¹⁸ at 450°C."

Table S1 Comparison of ASR values between SCNT and other highly active cathode compositions in literatures

Cathode Material Compositions	Area specific resistance ($\Omega \text{ cm}^2$)	Reference
SrCo _{0.8} Nb _{0.1} Ta _{0.1} O _{3-δ}	~0.16 @500°C ~0.68 @450°C	This work
Ba ₂ Bi _{0.1} Sc _{0.2} Co _{1.7} O _{6-δ}	~1.50 @ 500°C	3
SrSc _{0.175} Nb _{0.025} Co _{0.8} O _{3-δ}	~0.32 @500°C	4
NdBa _{0.75} Ca _{0.25} Co _{0.25} Co ₂ O _{5+δ}	~0.67 @500°C	5
Ba _{0.9} Co _{0.7} Fe _{0.2} Mo _{0.1} O _{3-δ}	~0.28 @500°C ~1.09 @450°C	6
Ba _{0.5} Sr _{0.5} Co _{0.8} Fe _{0.2} O _{3-δ}	~0.50 @500°C	7
SrSc _{0.2} Co _{0.8} O _{3-δ}	~0.45 @500°C	8

The authors carried out the first-principles calculations to support the synergetic effects of Nb and Ta co-doping. However, I notice several critical problems in the computational results and ensuing analysis, as follows:

1. The vertical scale in Figure 4(c) is different from those in Figure 4(a) and (b). This makes the enhancement of DOS at the Fermi level difficult to capture from the figure. The authors should use the same scale and provide the detailed number.

Reply:

We really appreciate Reviewer #2's meticulous observation. Accordingly, we have redrawn the figures using the same scale, and presented detailed comparisons of DOS near the Fermi level on Co of the three models.

2. The authors projected DOS on the Co atoms nearest to dopants. However, there are different types among the nearest Co atoms. For example, in Figure 4(a), dopant-coordination numbers of Co atoms are 0, 2, and 4. Which atoms are used in Figure 4(a) (and also (b) and (c))?

Reply:

For Nb-, Ta- and Nb/Ta-doped models, each dopant has six neighbouring Co atoms, which belong to two types: one is coordinated by four dopants, and another is coordinated by two dopants. The projected DOS on these nearest neighbouring Co atoms to dopants show similar trend, so we presented the data of the later case in the nearest neighbouring Co PDOS in the newly updated Figure 4(a)(d)(g).

3. In addition, non-neighbouring Co atoms should be also discussed in terms of the change in PDOS because they can also be the catalytic sites when exposed at the surface.

Reply:

Thanks for Reviewer #2's invaluable advice. We therefore considered the DOS of the Co atoms next nearest neighbour (NNN) to the dopants and observed that the DOS of NNN Co near the Fermi level is higher for Nb-doped model than Ta-doped model. Please see the updated Figure 4(b)(e)(h) in the main text. For PDOS on NNN Co, the Nb/Ta co-doped model has ~ 98 % DOS of the Nb-doped one near the Fermi level, while Ta-doped model has only 60%, implying that doping Nb into Ta-doped system can make NNN Co more active for electron transfer, which is beneficial for charge transfer process. This finding can also explain the more efficient charge process on SCNT as compared to SCT20 in spite of their similar level of oxygen vacancy content. Accordingly, we added the discussions in the main text as follows:

“As shown in the schematic models (Figure 4(c), (f) for Nb or Ta single-doped models and Figure 4(i) for co-doped model), there are two categories of cobalt atoms: one is the nearest neighbour (NN) Co to the dopant, including Co1, Co2 for single-doped model and Co1, Co2, Co3 for co-doped model; the other is the next nearest neighbour (NNN) Co to the dopants, including Co3 for single-doped model and Co4 for co-doped one. By comparing the electronic states of these Co atoms, we found that the NN-Co atoms have very similar DOS near the Fermi level for these three models. For the NNN-Co atoms, Ta-doped model (Figure 4(e)) exhibits only 60% of DOS of Nb-doped model (Figure 4(b)) near the Fermi level, indicating that Nb is more favourable to increase the DOS of the NNN-Co near the Fermi level. Due to this beneficial effect from Nb, the DOS of NNN Co near Fermi level for co-doped model (Figure 4(h)) shows ~ 98% that of Nb-doped one. The enhanced DOS at Fermi level can make electron transfer more efficient,⁴⁹ and thereby contribute to an improved charge-transfer process. Therefore, it is likely that the higher DOS of NNN-Co atoms near Fermi level as induced by Nb is the reason for the faster kinetics of charge-transfer steps of SCNT than that of SCT20, in spite of their similar concentration of oxygen vacancies.”

4. Most importantly, the increase of DOS by co-doping is difficult to rationalize if oxidation states of Nb and Ta in SCNT are the same as in SCN or SCT. The authors should clarify why DOS in SCNT is not a linear interpolation between those in SCN and SCT. This should be possible with the detailed analysis on the electronic structure.

Reply:

We feel very grateful for Reviewer #2's invaluable advice. Firstly, we apologized for presenting a wrong DOS of SCNT in the previous version. One of the PDOS result is not the DOS of the Co nearest neighbour (NN) to dopants, but the DOS of next nearest neighbour (NNN) to the dopants. The statement of the increase of DOS by co-doping is invalid. From the revised Figure 4, we can see that the DOS of SCNT is in between the data of SCN and SCT, and close to that of the SCN. Therefore, in view of DOS, the SCNT possesses similar surface activity as the SCN for electron transfer which is higher than SCT, which can explain the higher kinetics of SCNT for charge-transfer process as compared to SCT20 despite their similar oxygen vacancies.

In order to interpret the origins of the high ORR performance of SCNT, we also calculated the formation energy and migration barrier of the oxygen vacancy defect (V_O) from first-principles. Both low migration barrier (high migration ability) and low V_O formation energy (high V_O density) are advantageous for improving ORR performance. It is found that the SCN has the lowest migration barrier but the highest V_O formation energy. The SCT, however, has the highest migration barrier but the lowest V_O formation energy. The calculation results are consistent with our experimental results. We deduce that the high ORR performance of the SCNT is the result of the good balance between the V_O density and migration ability, as well as the high surface activity. Accordingly, we added more discussions in the "synergistic effects of Nb and Ta on the ORR" section of the main text as follows and a Table S2 in the Supplementary Information:

"The oxygen vacancy content of SCN20, SCT20 and SCNT as determined from NPD refinement at room temperature is 0.102 ± 0.02 , 0.159 ± 0.15 and 0.168 ± 0.15 , respectively, reflecting that SCNT and SCT20 have similar oxygen vacancy contents, which are both significantly higher than that of SCN20. Thermal gravimetric analysis also shows higher oxygen vacancy contents in SCNT and SCT20 than that of SCN20 at elevated temperature. (Supplementary Information Figure S9) Given the fixed valence of dopants, the valence of reducible Co is likely the main reason for oxygen vacancy concentration difference, so we calculated average valence of cobalt of samples from element contents as determined by the refinement. The average valence of Co is 3.44, 3.33 and 3.41 for SCN20, SCT20 and SCNT, respectively. The lower Co valence in Ta-doped samples can be ascribed to the lower electronegativity of Ta than Nb.⁴⁵ In addition, our first-principles calculation result also show that oxygen formation energy are 1.539 eV, 1.456 eV, and 1.512 eV for the Nb-, Ta-, and co-doped models, respectively, which further supports the observed higher oxygen deficiency in SCNT as induced by Ta."

"In order to confirm this hypothesis, we investigated the pathways for an oxygen vacancy migration through first-principles calculations. It is found that the three models have the same minimum energy pathways, as shown in Figure 3(b), but different energy barriers. The highest energy barriers along the pathway are 0.433 eV, 0.638 eV, and 0.572 eV for Nb-, Ta-, and co-doped models, respectively, (Supplementary Information Table S2), indicating a higher vacancy mobility of co-doped model as compared to Ta-doped one. Although SCNT and SCT20 have similar oxygen vacancy levels, the higher ionic conductivity of SCNT than SCT20 is likely a result of the incorporation of Nb dopant that can enhance ionic mobility in the lattice."

5. Related to 4, magnetic moments on Nb and Ta sites should be also analyzed if they are not in 5+ state.

Reply:

Thanks for the kind advice. We conducted XPS analysis on SCNT sample, and found that Nb and Ta are both in 5+ valence states, which is the same for dopants in SCN20 and SCT20 according to our previous work.⁹ We provided the XPS results in the supplementary information Fig. S2.

6. The computational section lacks in the detailed information on the computation (k-point mesh, U values, etc.)

Reply:

We appreciate Reviewer #2's careful observation. Correspondingly, the k-point mesh used for Brillouin zone integration and the U values are given in the experimental section as follows:

"...where the coulomb (U) and exchange (J) parameters are combined into the single parameter U_{eff} ($U_{eff} = U - J$) which was set to 0.8 eV in these calculations..."

"...The Brillouin zone was sampled using a $3 \times 3 \times 3$ k-point grid. The formation energy of an oxygen vacancy was calculated from the energy difference between the total energy of the V_O -containing sample and sum of the total energy of pristine sample and the chemical potential of an oxygen atom in an O_2 molecule. The minimum energy pathway for V_O migration was determined using a climbing image nudged band method^{61,62} implemented in the VASP code"

Reviewer #3

Manuscript presents a new high performance (low ASR) SCNT cathode for low temperature SOFC operation. This is technology that can have major impact and as such is appropriate for this journal. However, some details cause concern and require significant clarification:

The ASR values were determined by EIS on SDC pellets using Ag current collectors. The ASR values are very low, which is the point of paper but it raises concern over signal to noise. At 550°C (Fig 2 c) non-ohmic part is $\sim 0.06 \text{ Ohm cm}^2$, but total is 3.4 Ohm cm^2 . How accurate were the relative measurements? What was pellet thickness and does the high frequency part correspond to the pellet ASR, or is there an additional Ohmic contribution to the cathode ASR?

Reply:

We really appreciate Reviewer #3's immense time and invaluable assistance, which significantly improves the quality and accuracy of our manuscript.

Reviewer #3 made a meticulous observation here. The area specific resistance is actually the intercept difference of the impedance spectra with the real axis. The electrolyte pellet thickness is $\sim 0.65 \text{ mm}$, but varies slightly for different samples. All the cathodes under investigation in this work have been tested using the same EIS method at least three times to ensure their accuracy.

For example, the following is the impedance spectra of SCNT cathode on SDC electrolyte with three different electrolyte thicknesses at 500 °C. They all exhibit similar polarization resistance in spite of their different ohmic resistances contributed by the electrolyte, proving that the thickness of the electrolyte has no significant effect on the ASR measurement.

Figure 1 Impedance spectra of SCNT cathode based on SDC electrolyte with different thicknesses at 500°C.

The intercept of the impedance spectra with the real axis at high frequency corresponds to the ohmic resistance contributed by the thick electrolyte pellet, which has far lower electronic conductivity as compared to cathode materials¹⁰. Therefore, this method to determine cathode ASR should be sufficiently accurate, which is a gold standard method to measure the cathodes' ASRs for SOFCs.^{4, 6,}

ASR depends not only on cathode composition, but also thickness, microstructure, etc. The cathode is supposedly only 10 micron thick which is too thin for real SOFC resulting in large sheet resistance. Ag paste was used to address this for button cell, but then also contributes to cathode performance. How thick was the Ag current collector, and did any of the Ag enter the cathode pores? Fig 3 SEMs don't show Ag coating.

Reply:

Reviewer #3 is correct that there are many factors that may affect ASR values. Therefore, we fabricated the cathodes following the similar procedure to ensure the microstructural similarity of the cathodes.

In order to study the effect of cathode thickness on its performance, we measured the ASR values of SCNT cathode with different cathode thicknesses as follows according to Reviewer #3's comments. The ASR decreases with the cathode thickness until $\sim 10 \mu\text{m}$, and then remains similar when the cathode thickness increases from $\sim 10 \mu\text{m}$ to $\sim 15 \mu\text{m}$. This result suggests that the utilization length, characterizing the size of a cathode active region for ORR, should be less than $10 \mu\text{m}$. Such utilization length is reasonable because the utilization lengths for normal MIEC cathodes are typically around 3-5 μm .¹¹ Consequently, we added the following figure into Figure S5.

Therefore, we believe that the cathode with $\sim 10 \mu\text{m}$ thickness is sufficient for catalyzing the oxygen reduction reaction, and is adequate to reflect its ORR activity. In addition, many literature studies also reported the ASRs of their samples based on $\sim 10 \mu\text{m}$ thick cathodes.^{4, 6, 7}

Additionally, the SCNT cathode exhibits the best ORR catalyzing performance of all cathodes with similar microstructure and cathode thickness under our investigations, and is especially better than the SCN20 and SCT20 having similar lattice geometry with SCNT. By ruling out the effects of microstructure and thickness, it is obvious that the synergistic effects of co-doping Nb and Ta plays a significant role for the ORR activity.

Figure 2 ASRs of SCNT cathode with different cathode thicknesses at 500°C.

The silver paste was used to offer the electrons to the cathode. It has been reported that the ASR is very large ($\text{ASR} = \sim 0.92 \Omega \text{ cm}^2$ at 750°C)² if silver is solely applied as the electrode for ORR⁷.

Therefore, the very low ASR of SCNT cathode should rise from SCNT novel composition. The following SEM image is an example of a SCNT cathode after EIS measurement with the silver paste. The silver paste thickness is $\sim 5 \mu\text{m}$, and the silver grain size is $> 3 \mu\text{m}$. It seems to be very difficult for silver to diffuse into the cathode pores under open circuit conditions, and there is no silver shown in the SEM image. Accordingly, we updated Figure S4 for illustration.

Figure 3 SEM image of cross-section of SCNT cathode on electrolyte.

The discussion states a synergy between Nb and Ta on the ORR, but how that is done is not shown. It assumes HF arc due to charge transfer and LF arc due to non-charge transfer, but also not shown. If assumption state so.

Reply:

We feel grateful for Reivewer #3's meticulous observation. We appologize for this mistake, and we have added the corresponding descriptions in the experiment sector:

“The mechanism of the SCNT ORR was studied by fitting the EIS impedance spectra at different pO_2 to the $R_e (R_1CPE_1) (R_2CPE_2)$ equivalent circuit model by using the LEVM software. R_e represents the ohmic resistance of the electrolyte; (R_1CPE_1) and (R_2CPE_2) stand for the two ORR processes at high frequency and low frequency respectively. The physical meaning of the ORR processes are determined by a parameter m given as follows¹²:

$$\frac{1}{R_p} \propto P_{O_2}^m$$

R_p is the polarisation resistance of the corresponding ORR process”

Moreover, it is based on our previous work¹ that the arc at HF corresponds to charge transfer and the one at LF to non-charge transfer for SCN20 and SCT20. Therefore, we assumed that SCNT may follow the same mechanism. In order to confirm this assumption, we studied the ASR of an SCNT cathode as a function of oxygen partial pressure from 550°C to 450°C. The results are presented in the Supplementary Fig. S13 as shown in below. The m value for the process at HF is very close to 0.25, and the one at LF is close to 0.5. Therefore, we confirmed that the process at HF is related to the

charge-transfer process, and the LF one is to non-charge-transfer process, which is similar to SCN20 and SCT20.

Figure 4 ASR values of cathodes at (a) high frequencies and (b) low frequencies in a function of oxygen partial pressure at different temperatures.

Since Nb5+ and Ta5+ are fixed valent, how would they contribute to ORR? DOS projections indicate that by co-doping they impose a greater effect on Co, but fact that they substituted for Co means there is less Co if as expect Co is the active cation in ORR.

Reply:

Firstly, we revised the first principle calculations according to Reviewer #2's kind advice. We found that the superior electroactivity of SCNT is a result of optimised balance of high oxygen vacancy content, high ionic mobility and surface electron transfer ability brought by co-doping Nb and Ta, which both have impacts on their neighbouring Co ions.

Secondly, we believe that the electroactivity depends not only on the Co content but also on the "absolute activity" of Co, which is affected by these dopants (Ta and/or Nb). Therefore, an optimised dopant level should exist to achieve both sufficient Co content and sufficiently high Co "absolute activity" as induced by dopants. In our case, SCNT with 20 mol% doping seems to be better than the 10% mol doping analogues. For example, Figure 2(a) shows a comparison of ASRs among SCNT, SCT10 and SCN10, with the latter two possessing higher content of Co ions (90 mol%) than SCNT (80 mol%). However, SCNT with less Co is more active than SCN10 and SCT10 with more Co, with an ASR of $\sim 0.16 \Omega \cdot \text{cm}^2$ for SCNT but ~ 0.48 and $\sim 0.35 \Omega \cdot \text{cm}^2$ for the latter two cathodes respectively at 500°C. The remarkable ORR activity enhancement, we think, is a result of the synergistic co-doping effects on cobalt, which significantly enhance the absolute activity of active sites by facilitating faster ionic bulk conduction and optimized surface charge transfer both confirmed in our work.

Thirdly, we think it is meaningful to know the optimised content for co-doping level, so we will further study effects of co-doping level on cathode electroactivity in our next work.

One of the most important Fig's is S8 which is in Supplemental rather than main text. It shows the conductivity of SCNT compared to the other single doped compounds SCT20 and SCN20. This 4pt measurement should have less error than the EIS and it shows SCNT is LESS conductive than SCT20 and SCN20 in the temperature range of interest, 400-550°C. If SCNT is the better cathode why does it have lower conductivity? I assume answer will be because that is electronic and not ionic conductivity, but then you need ionic conductivity measurements in this temperature region.

O₂ permeation measurements were done to separate out ionic conductivity and it shows higher conductivity for SCNT, but only over 700-860 °C temp range which is not the temperature range of the ASR and SOFC measurements. Also these types of measurements are prone to gas leaks. Did they use a tracer gas to determine any leak rate

Reply:

Thanks for Reviewer #3's suggestion. Fig. S10 shows the electrical conductivity of the samples, and electrical conductivity consists of electronic and ionic conductivity. However, the former dominates the electrical conductivity because of its significantly larger value than the latter. Therefore, SCNT indeed has lower electronic conductivity than SCN20 and SCT20, but with a small disparity less than 13%. The low electrical conductivity is likely a result of SCNT's high content of oxygen vacancies, which can diminish charge carriers for hopping. Because the electronic conductivity is shown to be a less important factor contributing to the good cathode performance of SCNT, it seems to be more suitable to put the electrical conductivity data in the supplementary information. Accordingly, we added discussions in the maintext:

"...The lower electrical conductivity is likely caused by more oxygen vacancies in SCNT that can diminish the charge carriers for hopping process..."

We agree with Reviewer #3 to provide the ionic conductivity comparison within a lower temperature range. Therefore, we reconducted the oxygen permeability tests, and then replaced Fig. 3 with a new one (as shown below) that presents a comparison of ionic conductivity among SCNT, SCN20 and SCT20 at 475°C~600°C. It clearly shows that SCNT has a higher level of ionic conductivity relative to SCN20 and SCT20. Moreover, electronic conductivity is reported to be a less important factor than ionic conductivity influencing the ORR activity.^{13, 14, 15, 16} For instance, the electronic conductivity of the benchmark cathode Ba_{0.5}Sr_{0.5}Co_{0.8}Fe_{0.2}O_{3-δ} is much lower than La_{0.6}Sr_{0.4}Co_{0.2}Fe_{0.8}O_{3-δ} (LSCF)¹⁵, but BSCF⁷ is more active than LSCF¹⁶ in catalyzing ORR at lower temperature.

In the gas permeability measurement, N₂ in the air was used as the tracer gas to determine the leak rate. The leak rate is ~ 0.5-0.7%. Moreover, we use the following equation (also shown in Experimental section) to determine the overall resistance to the oxygen permeation, which removes the influence of slight leaks on the ionic conductivity. Therefore, this method is able to estimate the samples' ionic conductivity accurately.

$$R_{overall} = \frac{RT}{16F^2} \frac{1}{Sj_{O_2}} \left[\ln \left(\frac{P'_{O_2}}{P''_{O_2}} \right) \right]$$

Figure 5 Estimated ionic conductivities of SCN20, SCT20 and SCNT from 600 °C to 475 °C.

Minor points:

GDC is Gadolinia Doped Ceria, which is "Gd" but authors keep using "Ga" which is Gallium. Pg 3 reference 11 is cited for their LSM-ESB cathode having high performance below 600 °C and authors state this is "result of enhanced ionic conduction". However, the ref 11 authors themselves describe this in terms of enhanced ORR due to LSM having excellent O₂ dissociation and ESB having excellent O incorporation steps. This should be more accurately reflected in current manuscript as well as including the performance itself which was 0.078 Ohm cm² at 600°C which is comparable to the following ref 12 performance.

Reply:

We apologize for this error, and we have corrected the GDC errors in the main text. What is more, we modified the issue in the introduction part as follows:

“For example, the in-situ co-assembly of La_{0.8}Sr_{0.2}MnO₃ (with a very low O₂ dissociative energy barrier) and Bi_{1.6}Er_{0.4}O₃ (with fast oxygen incorporation kinetics) leads to a high performance nanocomposite cathode with a low polarisation resistance of ~0.078 Ω cm² at 600 °C.¹¹”

Fig 5a shows an increase in ASR with time. What is the % change per hr?

Reply:

We appreciate Reviewer #3’s careful observation. The ASR change increases by ~0.06% per hour during the testing period. The increase may arise from the change of microstructure of Ag current collector at 600°C, which has been reported before.^{2,17} Correspondingly, we made some modifications in the main article as follows:

“The ORR activity was relatively stable at ~ 0.033 Ω·cm² with an ASR increase of ~0.06% /h during the testing period. The slight increase of ASR during the stability test is likely to arise from the densification and the reduced porosity of the silver current collector, which degrades the overall cathode performance during this testing timeframe.^{50,51,52}”

Fig S7 identify the layers. Also, (b) is poor resolution image with charging

Reply:

Thank you for Reviewer #3's meticulous observation, and we have replaced the poor resolution one with one without charging in the supplementary information Fig. S8, and the corresponding layers are also identified.

References

1. Li M, Zhou W, Peterson VK, Zhao M, Zhu Z. A comparative study of $\text{SrCo}_{0.8}\text{Nb}_{0.2}\text{O}_{3-\delta}$ and $\text{SrCo}_{0.8}\text{Ta}_{0.2}\text{O}_{3-\delta}$ as low-temperature solid oxide fuel cell cathodes: effect of non-geometry factors on the oxygen reduction reaction. *Journal of Materials Chemistry A* **3**, 24064-24070 (2015).
2. Camaratta M, Wachsman E. Silver–bismuth oxide cathodes for IT-SOFCs; Part I — Microstructural instability. *Solid State Ionics* **178**, 1242-1247 (2007).
3. Zhou W, *et al.* Novel B-site ordered double perovskite $\text{Ba}_2\text{Bi}_{0.1}\text{Sc}_{0.2}\text{Co}_{1.7}\text{O}_{6-x}$ for highly efficient oxygen reduction reaction. *Energy & Environmental Science* **4**, 872-875 (2011).
4. Zhou W, Sunarso J, Zhao M, Liang F, Klande T, Feldhoff A. A Highly Active Perovskite Electrode for the Oxygen Reduction Reaction Below 600 °C. *Angewandte Chemie International Edition* **52**, 14036-14040 (2013).
5. Yoo S, *et al.* Development of Double-Perovskite Compounds as Cathode Materials for Low-Temperature Solid Oxide Fuel Cells. *Angewandte Chemie International Edition* **53**, 13064-13067 (2014).
6. Huang S, Lu Q, Feng S, Li G, Wang C. $\text{Ba}_{0.9}\text{Co}_{0.7}\text{Fe}_{0.2}\text{Mo}_{0.1}\text{O}_{3-\delta}$: A Promising Single-Phase Cathode for Low Temperature Solid Oxide Fuel Cells. *Advanced Energy Materials* **1**, 1094-1096 (2011).
7. Shao Z, Haile SM. A High-Performance Cathode for the Next Generation of Solid-Oxide Fuel Cells. *Nature* **431**, 170-173 (2004).
8. Zhou W, Shao Z, Ran R, Cai R. Novel $\text{SrSc}_{0.2}\text{Co}_{0.8}\text{O}_{3-\delta}$ as a Cathode Material for Low Temperature Solid-Oxide Fuel Cell. *Electrochemistry Communications* **10**, 1647-1651 (2008).
9. Li M, Zhou W, Peterson VK, Zhao M, Zhu Z. A Comparative Study of $\text{SrCo}_{0.8}\text{Nb}_{0.2}\text{O}_{3-\delta}$ and $\text{SrCo}_{0.8}\text{Ta}_{0.2}\text{O}_{3-\delta}$ as Low-Temperature Solid Oxide Fuel Cell Cathodes: Effect of Non-Geometry Factors on the Oxygen Reduction Reaction. *Journal of Materials Chemistry A*, (2015).
10. Steele BCH, Heinzel A. Materials for fuel-cell technologies. *Nature* **414**, 345-352 (2001).
11. Adler SB. Factors Governing Oxygen Reduction in Solid Oxide Fuel Cell Cathodes. *Chemical Reviews* **104**, 4791-4844 (2004).
12. Takeda Y, Kanno R, Noda M, Tomida Y, Yamamoto O. Cathodic polarization phenomena of perovskite oxide electrodes with stabilized zirconia. *Journal of The Electrochemical Society* **134**, 2656-2661 (1987).

13. Wang L, Merkle R, Mastrikov YA, Kotomin EA, Maier J. Oxygen Exchange Kinetics on Solid Oxide Fuel Cell Cathode Materials—General Trends and Their Mechanistic Interpretation. *J Mater Res* **27**, 2000-2008 (2012).
14. Wang L, Merkle R, Maier J. Surface Kinetics and Mechanism of Oxygen Incorporation Into $\text{Ba}_{1-x}\text{Sr}_x\text{Co}_y\text{Fe}_{1-y}\text{O}_{3-\delta}$ SOFC Microelectrodes. *J Electrochem Soc* **157**, B1802-B1808 (2010).
15. Jun A, Yoo S, Gwon O-h, Shin J, Kim G. Thermodynamic and electrical properties of $\text{Ba}_{0.5}\text{Sr}_{0.5}\text{Co}_{0.8}\text{Fe}_{0.2}\text{O}_{3-\delta}$ and $\text{La}_{0.6}\text{Sr}_{0.4}\text{Co}_{0.2}\text{Fe}_{0.8}\text{O}_{3-\delta}$ for intermediate-temperature solid oxide fuel cells. *Electrochimica Acta* **89**, 372-376 (2013).
16. Perry Murray E, Sever MJ, Barnett SA. Electrochemical performance of (La,Sr)(Co,Fe)O₃–(Ce,Gd)O₃ composite cathodes. *Solid State Ionics* **148**, 27-34 (2002).
17. Chen Y, *et al.* Role of silver current collector on the operational stability of selected cobalt-containing oxide electrodes for oxygen reduction reaction. *Journal of Power Sources* **210**, 146-153 (2012).

Reviewers' comments:

Reviewer #1 (Remarks to the Author):

I believe that the current revised manuscript is good shape to be published in this journal. I recommend this manuscript to Nature Communications.

Reviewer #2 (Remarks to the Author):

The authors improved the computational part significantly by responding to my criticisms properly. I have just one minor comment: in Fig. 3(b), the numbers 1,..,5 not explained neither in the caption nor in the main text. Besides this, I recommend its publication.

Reviewer #3 (Remarks to the Author):

The authors have made a good attempt to answer most of my previous questions, but some remain:

1. With regard to EIS ASR, my comment was that because ASR was so low it was possible there was overlap in the electrolyte and electrode ohmic contributions. In this regard it would be worthwhile to calculate the electrolyte ASR based on known conductivity and thickness and subtract from the high frequency intercept to see if there is a differential (positive or negative) that would be ascribed to the electrode ohmic ASR. Instead authors provided new figure with 3 cells of different SDC thickness to show data was reproducible. This data then brings up question why does cell 3 have lower non-ohmic polarization than the other 2? Again, would be worthwhile to subtract GDC ASR from these additional cells to determine if there is an ohmic electrode contribution.

2. I asked about evidence ascribing HF arc to charge transfer and LF to non-charge transfer and authors used PO₂ dependence to show HF had 1/4 dependency and LF had 1/2 dependency and then stated this "confirmed" this assignment. This PO₂ dependence does not "confirm" this assignment, but it is "consistent with" others who have claimed this dependency.

3. I was glad see the new figure of ionic conductivity (Fig 3a) in the temperature range of interest in response to my previous question. This clearly shows SCNT has higher ionic conductivity than SCN and SCT and makes more sense with regard to the corresponding electrode EIS derived ASR results than the electronic conductivity measurements which showed opposite trend. However, since this is supposed to be an MIEC electrode material authors should then calculate and present ionic transference (t_i) data. In this regard I see a 500°C ionic conductivity of 0.005 S/cm (Fig 3a) and an electronic conductivity of 135 S/cm (Fig S10), that corresponds to only a t_i of 0.00004 which is extremely small and indicates the material has negligible ionic conductivity.

4. The addition of the comparison table S1 was a good addition. However, I would add the values from the rest of the ref's (11, 12, 7 and 18) in the introduction section. For example reading off the figure in ref 11 the ASR at 500°C is same as BSCF (~0.5 S/cm).

Reviewer #1

I believe that the current revised manuscript is good shape to be published in this journal. I recommend this manuscript to Nature Communications.

Reply: We appreciate the invaluable advice and support of Reviewer #1.

Reviewer #2

The authors improved the computational part significantly by responding to my criticisms properly. I have just one minor comment: in Fig. 3(b), the numbers 1,...5 not explained neither in the caption nor in the main text. Besides this, I recommend its publication.

Reply:

We are grateful to Reviewer #2 for their meticulous examination of our work. Accordingly, we added a sentence in the caption of Fig.3 to explain these numbers: “*The numbers 1-5 indicate the sequential positions of oxygen vacancies along the diffusion pathway.*” We thank Reviewer #2 for their review, which has enhanced the accuracy and quality of this manuscript.

Reviewer #3

The authors have made a good attempt to answer most of my previous questions, but some remain: 1. With regard to EIS ASR, my comment was that because ASR was so low it was possible there was overlap in the electrolyte and electrode ohmic contributions. In this regard it would be worthwhile to calculate the electrolyte ASR based on known conductivity and thickness and subtract from the high frequency intercept to see if there is a differential (positive or negative) that would be ascribed to the electrode ohmic ASR. Instead authors provided new figure with 3 cells of different SDC thickness to show data was reproducible. This data then brings up question why does cell 3 have lower non-ohmic polarization than the other 2? Again, would be worthwhile to subtract GDC ASR from these additional cells to determine if there is an ohmic electrode contribution.

Reply:

Many thanks to Reviewer #3’s invaluable advice and contribution to this review, which has greatly enhanced the quality of our work.

Figure 1 A comparison of ohmic resistances of a GDC-based symmetrical cell with SCNT electrode and a GDC-based symmetrical cell with silver electrode.

In order to answer this question, we fabricated another GDC-based symmetrical cell with silver electrodes by applying silver paste to both sides of a GDC pellet followed by baking at 260 °C. The thickness of the electrolyte is the same as that of the symmetrical cell with SCNT electrode. For the symmetrical cell with silver electrode, due to the very high electrical conductivity of silver, the intercept of impedance at high frequencies can be estimated to be the ohmic resistance of the GDC electrolyte. We compared the high-frequency intercepts of the two symmetrical cells with silver and SCNT as electrode, respectively, between 450-700 °C using EIS, and the results are presented in the Figure 1 above. The ohmic resistance of the symmetrical cell with the SCNT electrode is higher by only ~1-2% than that of the symmetrical cell with the silver electrode, which means that the SCNT cathode on both sides of GDC contributes only about 1-2% to the whole ohmic resistance. Therefore, Reviewer #3 is right that there are cathode contributions to the ohmic resistances, but we show by experiments that the contributions are very small in the symmetrical cell. Accordingly, we have added this figure into the supporting information (Fig. S2(a)) of the revised work alongside a comment in the main text:

“...with low ASR indicating high activity. The intercept of the impedance at high frequencies indicates an ohmic resistance arising from the electrolyte, electrode, and connection wires,⁴⁰ with only ~1-2% of the total ohmic resistance arising from the SCNT cathode on both sides of the electrolyte GDC-based symmetrical cell (Supplementary Fig. S2(a)).”

2. I asked about evidence ascribing HF arc to charge transfer and LF to non-charge transfer and authors used PO₂ dependence to show HF had 1/4 dependency and LF had 1/2 dependency and then stated this "confirmed" this assignment. This PO₂ dependence does not "confirm" this assignment, but it is "consistent with" others who have claimed this dependency.

Reply:

Reviewer #3 is right that our observation is merely consistent with other researchers regarding the relations between rate controlling steps and the oxygen partial-pressure dependence of the cathode reciprocal resistance. Accordingly, we corrected the main text:

“As shown in Supplementary Fig. S12 and from our previous work⁴⁴, the cathode reciprocal resistances at high and low frequencies show different oxygen partial pressure dependencies, m (as shown in Eq 5): m is ~1/4 at high frequencies (HF) and ~1/2 at low frequencies (LF). According to the relationship between pO_2 dependencies and rate-determining-steps as discussed by other researchers,^{50,51,52} the polarisation resistance at HF is likely related to the charge transfer and that at LF to non-charge-transfer steps.”

3. I was glad see the new figure of ionic conductivity (Fig 3a) in the temperature range of interest in response to my previous question. This clearly shows SCNT has higher ionic conductivity than SCN and SCT and makes more sense with regard to the corresponding electrode EIS derived ASR results than the electronic conductivity measurements which showed opposite trend. However, since this is supposed to be an MIEC electrode material authors should then calculate and present ionic transference (t_i) data. In this regard I see a 500°C ionic conductivity of 0.005 S/cm (Fig 3a) and an electronic conductivity of 135 S/cm (Fig S10), that corresponds to only a t_i of 0.00004 which is extremely small and indicates the material has negligible ionic conductivity.

Reply:

Many thanks to Reviewer #3's helpful suggestions. As per Reviewer #3's comments, we calculated and compared the ionic transference numbers (t_{ion}) of SCNT, SCN20, and SCT20 samples. We found that SCNT exhibits the highest t_{ion} amongst the samples, suggesting that it is the co-doping that allows more rapid oxygen-ion conduction. We have included a new figure showing these transference numbers in Fig. S10 and added discussions in the main text:

“Additionally, slightly lower electrical conductivity (σ_{total}) that includes both electronic and ionic conductivity (σ_{ion}) is observed for SCNT than for SCN20 and SCT20 (Supplementary Fig. S9). The lower electrical conductivity of SCNT is caused by increased oxygen vacancies that can diminish the charge carriers for hopping process. The ionic transference number ($t_{ion} = \sigma_{ion}/\sigma_{total}$) of SCNT, SCN20, and SCT20 was calculated and shown in Figure S10. As the electronic conductivity dominates, the ionic transference number is very small in the temperature range studied. Nevertheless, SCNT has a larger ionic transference than SCN20 and SCT20; e.g. SCNT has a transference number of $\sim 1.33 \times 10^{-5}$ at 500 °C which is ~ 2.7 and ~ 2.1 times that of SCN20 and SCT20, respectively.”

4.The addition of the comparison table S1 was a good addition. However, I would add the values from the rest of the ref's (11, 12, 7 and 18) in the introduction section. For example reading off the figure in ref 11 the ASR at 500°C is same as BSCF (~ 0.5 S/cm).

Reply:

We agree with Reviewer #3's kind suggestion, and have included these data in Table S1 accordingly.

REVIEWERS' COMMENTS:

Reviewer #3 could not be reached for final comments.